# The haplo-diplontic life cycle expands niche space of coccolithophores

Joost de Vries[1, 2], Fanny Monteiro[1], Glen Wheeler[2], Alex Poulton[3], Jelena Godrijan[4], Federica Cerino[5], Elisa Malinverno[6, 7], Gerald Langer[2], and Colin Brownlee[2, 8]

[1]BRIDGE, School of Geographical Sciences, University of Bristol, University Road, Bristol BS8 1SS, UK
[2]Marine Biological Association, The Laboratory, Citadel Hill, Plymouth PL1 2PB, UK
[3]The Lyell Centre for Earth Marine Science Technology, Heriot-Watt University, Edinburgh, UK
[4]Division for Marine and Environmental Research, Ruđer Bošković Institute, Bijenička cesta 54, 10000 Zagreb, Croatia
[5]Oceanography Section, Istituto Nazionale di Oceanografia e di Geofisica Sperimentale - OGS, via Piccard 54, 34151 Trieste, Italy
[6]University of Milano-Bicocca, Department of Earth and Environmenal Sciences, Piazza della Scienza, 4 - 20126 Milano, Italy
[7]Consorzio Nazionale Interuniversitario per le Scienze del Mare - CoNISMa, Piazzale Flaminio 9 - 00196 Roma, Italy
[8]School of Ocean and Earth Science, University of Southampton, Southampton SO14 3ZH, UK

**Correspondence:** Joost de Vries (joost.devries@bristol.ac.uk)

**Abstract.** Coccolithophores are globally important marine calcifying phytoplankton that utilize a haplo-diplontic life cycle. The haplo-diplontic life cycle allows coccolithophores to divide in both life cycle phases, and has been proposed to allow coccolithophores to expand their niche space. To-date research has, however, largely overlooked the life cycle of coccolithophores, and instead focused on the diploid life cycle phase of coccolithophores. Through the synthesis and analysis of global scanning electron microscopy (SEM) coccolithophore abundance data (n = 2534), we find that calcified haploid coccolithophores generally constitute a minor component of the total coccolithophore abundance ($\approx 2 - 15\%$ depending on season). However, using case studies in the Atlantic Ocean and Mediterranean Sea, we show that depending on environmental conditions calcifying haploid coccolithophores can can be significant contributors to the coccolithophore standing stock (up to $\approx 30\%$). Furthermore, using hypervolumes to quantify the niche space of coccolithophores, we illustrate that the haploid and diploid life cycle phases inhabit contrasting niches, and that on average, this allows coccolithophores to expand their niche space by $\approx$18.8 % with a range of 3-76% for individual species.

Our results highlight that future coccolithophore research should consider both life cycle stages, as omission of the haploid life cycle phase in current research limits our understanding of coccolithophore ecology. Our results furthermore suggest a different response to nutrient limitation and stratification, which may be of relevance for further climate scenarios.

Our compilation highlights the spatial and temporal sparsity of SEM measurements, and the need for new molecular techniques to identify uncalcified haploid coccolithophores. Our work also emphasizes the need for further work on the carbonate chemistry niche space of the coccolithophore life cycle.

# 1 Introduction

Coccolithophores are marine phytoplankton that produce calcium carbonate platelets, called 'coccoliths', which can be seen from space when coccolithophores bloom. Coccoliths eventually rain down into the ocean interior or serve as ballast as they are incorporated into faecal pellets and aggregates, which drives the carbonate pump and enhances the organic carbon pump by increasing organic carbon export rates to the deep sea (Klaas and Archer, 2002; Zeebe, 2012). Through the production of coccoliths, coccolithophores produce $\approx$1.5 Pg of inorganic carbon per year (Hopkins and Balch, 2018; Krumhardt et al., 2019), and subsequently account for 30 % to 90 % of carbonate in sediments (Broecker and Clark, 2009) highlighting the importance of coccolithophores in calcium carbonate burial..

In addition to the carbonate pump, coccolithophores contribute to the organic carbon pump, accounting for 1-40 % of marine primary production depending on habitat (Poulton et al., 2007, 2013). Because of involvement in the ocean carbon pumps and food web, coccolithophores thus play an important role in the ocean on regional to global spatial scales and seasonal to geological time scales.

Much focus has been put on understanding coccolithophore ecology and physiology, such as the function of calcification (Young, 1994; Monteiro et al., 2016; Xu et al., 2016), their diversity (Aubry, 2009; Young et al., 2003), the factors controlling their calcification (Zondervan, 2007; Taylor et al., 2017) and competitiveness (Margalef, 1978; Krumhardt et al., 2017). However, one factor that significantly impacts coccolithophore calcite production and potentially their global success, has been given little attention: their distinctive life cycle.

The life cycle of an organism is defined by the number of chromosome sets (the 'ploidy level') of the cell when asexual reproduction ('mitosis') occurs. If mitosis occurs when the cell has one set of chromosomes (a haploid cell) the life cycle is called 'haplontic' (Fig. 1a), while if mitosis occurs when the cell has two sets of chromosomes (a diploid cell) the life cycle is called 'diplontic' (Fig. 1b). A few organisms can divide in both the haploid and diploid phase. Such a life cycle is called 'haplo-diplontic' (Fig. 1c). Coccolithophores utilize the latter life cycle strategy - which is in contrast to dinoflagellates and diatoms which tend to be either haplontic or diplontic, and as such can only divide in either the haploid or diploid life cycle phase (Von Dassow and Montresor, 2011).

The haploid and diploid life cycle phases of coccolithophores can vary significantly in terms of coccolith structure, size and morphology, cell size, and degree of calcification (Fig. 2). The diploid life cycle phases tend to be more heavily calcified than the haploid life cycle phases, which tend to be more lightly, or non-calcified (Cros et al., 2000; Daniels et al., 2016; Fiorini et al., 2011a, b). This difference in cell calcium carbonate content, cell organic carbon content, and ratio thereof (the PIC:POC ratio), between the two life cycle phases means that the two phases potentially have contrasting impacts on the carbonate pump.

Although coccolithophore morphology is highly diverse, the diploid phases of coccolithophores primarily utilize hetero-coccolithophore morphology (with some exceptions, i.e. *Braarudosphaera bigelowii*), while the haploid life cycle phases can broadly be classified into four morphologies: polycrater (Fig. 2a), ceratolith (Fig. 2b), holococcolith (Fig. 2c-i) and unmineralized (not pictured) (Frada et al., 2018). Of these four haploid morphologies, the holococcolithophore morphology - which is defined by rhomboid calcite structures that constitute the coccoliths - is the most frequent utilized (Frada et al., 2018). Eight

coccolithophore clades utilize holococcoliths, while four clades utilize an unmineralized haploid morphology, one clade utilizes a ceratolith morphology, one clade utilizes ceratolith morphology, and for five clades the haploid morphology is currently unknown (Frada et al., 2018).

Coccolith and coccosphere morphology, cell and coccosphere size, and the degree of calcification influences coccolithophore ecology (Young, 1994). We can thus expect that the haploid and diploid life cycle phases of coccolithophores can have contrasting ecological preferences, which has been proposed to allow a coccolithophore species to occupy multiple niches (Houdan et al., 2006; Frada et al., 2012; Cros and Estrada, 2013; Godrijan et al., 2018; Frada et al., 2018). This ability to occupy multiple niches should expand the total niche space coccolithophore can inhabit, a potential advantage for haplo-diplontic organisms in

variable environments (Mable and Otto, 1998). An idea which is supported by genetic models (Hughes and Otto, 1999; Rescan et al., 2015).

While niche differentiation has been widely observed for haplo-diplontic seaweeds (Couceiro et al., 2015; Guillemin et al., 2013; Lees et al., 2018; Lubchenco and Cubit, 1980), and coccolithophores (Houdan et al., 2006; Cros and Estrada, 2013; Godrijan et al., 2018; Frada et al., 2018), to-date no research has quantitatively investigated the extent of niche overlap and

niche expansion for haplo-diplontic algae. For coccolithophores this is because research has primarily focused on the diploid life phases, and relatively little is known in regards to the haploid life phase (Taylor et al., 2017; Frada et al., 2018). This is in part due to a research focus on the globally ubiquitous *Emiliania huxleyi* which utilizes an unmineralized haploid morphology which cannot be readily identified with conventional light or scanning electron microscopy (Frada et al., 2008).

With the aim of understanding how haploid coccolithophores contribute to coccolithophore success, we quantify the niche

overlap and niche expansion between haploid and diploid life stages of coccolithophores for the first time.

To do so, we compile global coccolithophore abundance observations of coccolithophores using all available SEM measurements and where appropriate corresponding environmental measurements (temperature, salinity, DIN, phosphate, and silicate). Although our focus is on holococcolith forming clades rather than *E. huxleyi*, holococcolith forming clades include ecologically relevant species such as *Helicosphaera carteri* (Fig. 2e), *Coccolithus pelagicus* and *Calcidiscus leptoporus* (Fig. 2i)

which contribute more to the CaC0$_3$ flux to the deep ocean than *E. huxelyi* due to their larger coccolith and coccosphere size (Ziveri et al., 2007; Rigual Hernández et al., 2019).

In addition to niche overlap and niche expansion, we investigate the dataset to identify ecological preferences of holococcolith forming species, providing an updated picture on their global distribution, relative abundance, niche space, and environmental controls. This work provides key information to better understand how the haplo-diplontic life cycle contributes to

coccolithophore success.

## 2    Methods

### 2.1    Metadata compilation

Coccolithophore abundance measurements were compiled from 36 studies, constituting 2534 measurements, and representing all major oceans (Table 1). These studies utilized scanning electron microscopy (SEM) to enumerate or further identify coccol-

ithophores rather than soley relying on the more commonly utilized light or cross polarized microscopy which under-represents coccolithophore biodiversity (Godrijan et al., 2018), in particular holococcolithophores (Bollmann et al., 2002; Cerino et al., 2017). We used this data set to investigate global, and vertical distribution patterns of haploid and diploid coccolithophore life cycle phases, specifically focusing on holococcolith forming species. Since abundance data was manually compiled, our data set is not exhaustive. For instance, some SEM studies such as those by Okada and Honjo (1973); Honjo and Okada (1974); Reid (1980) are not included in this data set since the data were not retrievable from the original publications.

In addition to the global data set, we further investigated three case studies, in order to better understand specific drivers and differences between the life cycle phases: the Atlantic Meridional Transect (AMT), representative of mid-oligotrophic open ocean ecosystems, the long term time series at Bermuda (BATS), and two times series in a mesotrophic coastal ecosystem in the Adriatic Sea (the 'Mediterranean data set'). For the AMT study, we considered observations from 4 cruises, specifically AMT-12 (May-Jun 2003), AMT-14 (Apr-Jun 2004), AMT-15 (Sep-Oct 2004) and AMT-17 (Oct-Nov 2005) previously published by Poulton et al. (2017). For the BATS station we considered data published by Haidar and Thierstein (2001), which consists of approximately monthly observations between January 1991 to January 1994. For the Mediterranean study, we combine two time-series in the Adriatic Sea by Godrijan et al. (2018) and Cerino et al. (2017), between September 2008 to December 2009 and May 2011 to February 2013 at the RV-001 and C1-LTER stations respectively.

For the BATS, AMT and Mediterranean case studies, we additionally compiled temperature, salinity, and concentrations of DIN (nitrite + nitrate), phosphate, and silicate. For the AMT environmental variables were acquired from the British Oceanographic Data Centre (BODC). For the BATS environmental variables were acquired from the Bermuda Institute of Ocean Sciences (BIOS). For the Mediterranean study, day length was calculated using the MIT Skyfield package in Python. Other environmental variables such as turbulence, irradiance and pH might also impact coccolithophore distribution patterns, but we have not included them into our compilation because they are not available for all presented case studies.

All data was acquired from supplementary data, online databases, or if neither was available by contacting the authors directly. The data was manually checked for synonyms or misspellings of species names, and where appropriate cell abundances were converted to cells $l^{-1}$. All species, or genera if not identified to a species level, were labeled as either heterococcolithophore, holococcolithophore, or 'other', which includes polycrater, nanoliths, and unidentified species. For these categorizations we followed definitions from Cros and Fortuño (2002).

The species and environmental data were compiled in Python, and subsequently analysed in R (R Core Team, 2019). For all analysis we only considered samples within the top 200 m of the water column. To reduce the effects of seasonality, we binned the data into four main seasons, defined as Dec-Feb, Mar-May, Jun-Aug, Sep-Nov. We also calculated on a global scale and regional scale the mean of the observed abundances and estimated the highest observed abundances (the 'maximum abundance') for both hetero- and holococcolithophores and each season. For the mean abundance calculations the mean was calculated for each sample and then averaged. Finally, we tested the count data for a normal distribution using a Shapiro-Wilks test for each region and the global data set. Where the count distribution was found normal (all data), a 95% confidence interval was calculated.

**Sampling bias and cover of data set**

Our compilation contains sampling bias and is spatially and temporally incomplete. Temporally, there is bias towards the months Jun-Aug and Dec-Feb (29.28% and 30.59% of samples, respectively), with fewer samples in the inter-seasons. This temporal bias results from generally higher sampling effort in the Arctic Circle in Jun-Aug (8.43% of samples) and the Southern Ocean in Dec-Feb (13.15% of samples). Not coincidentally, this is when and where coccolithophore abundances are the highest (see results below). When excluding the Arctic Circle (Jun-Aug) and the Southern Ocean (Dec-Feb), the data set is temporally

relatively evenly distributed (28.20% Mar-May, 26.58% Jun-Aug, 22.13% Sep-Nov, 22.23% Dec-Feb).

Spatially, there is higher sampling in the Atlantic Ocean, Mediterranean Sea, Arctic Circle and Southern Ocean. In terms of spatial cover, coverage is limited in the Pacific Ocean and data is lacking in the Southern Ocean between Jun-Aug and the Arctic Circle between Dec-May. However, previous studies note the low coccolithophore abundance in the tropical and subtropical Pacific Ocean (Okada and Honjo, 1973; Honjo and Okada, 1974; Reid, 1980), and the absence or low abundance

of holococcolithophores in this region (Okada and Honjo, 1973; Honjo and Okada, 1974; Reid, 1980). The lack of data in the Southern Ocean and the Arctic Circle for specific months is due to the difficulty of sampling these regions in the winter as well as low coccolithophore abundance due to light limitation.

The incomplete data cover of our data set combined with the spatial and temporal bias means that the analysis presented here mainly serves as a first order estimate of the relative hetero- and holococcolithophore abundance and distribution patterns.

For more accurate estimates, additional sampling needs to be conducted.

For more absolute estimates, additional sampling will have to be conducted. Although inter-annual variability and strong links between coincident climate variability and primary productivity (Behrenfeld et al., 2006) as well as inter-annual and mesoscale variability on local scales will influence phytoplankton distribution patterns (Volpe et al., 2012) which makes estimating global abundances challenging.

**2.2 Definition of pairs and HOLP-index**

Not all heterococcolithophore forming coccolithophore species form holococcospheres. Thus, to better illustrate the proportion of haploid and diploid coccolithophore cells, we reported the ratio between hetero- and holococcospheres of species that form holococcoliths in their haploid phase, which is commonly implemented (Cros and Estrada, 2013; Šupraha et al., 2016).

This ratio is referred to as the 'HOLP-index', and is defined by Cros and Estrada (2013) as:

$$\text{HOLP-index} = 100 \cdot \frac{\text{paired holococcolithophore abundance}}{\text{paired coccolithophore abundance}} \qquad (1)$$

Species included in the HOLP-index follow the definitions of paired species as defined in Frada et al. (2018) (Table 2) - which is confined to currently understood associations and which is likely to change as our understanding holococcolith species continues to improve. We calculated the HOLP-index on a global and regional level for studies that identified holococcolithophores to a species level, the AMT data set, and the Mediterranean data set. To calculate the mean HOLP-index, the

150 ratios were calculated for each sample and then averaged.

## 2.3 Environmental drivers

We quantified the environmental drivers of hetero- and holococcolithophore abundance and the HOLP-index for the AMT and Mediterranean data sets using Spearman correlations. We calculated Spearman correlations for hetero- and holococcolithophores and the HOLP-index relative to temperature, salinity, depth, and concentrations of DIN (nitrite + nitrate), phosphate, and silicate for the AMT data set. The same ordinal associations were calculated for the Mediterranean data set, but we considered day length instead of depth, because only the top 30 meters of the water column was sampled, and seasonality is an important driver in this region. To focus on marine systems of coccolithophores, we only considered samples with salinities above 30 ppt. Samples missing any environmental variables were removed. Subsequently the AMT data set included a total of 45 samples, and the Mediterranean data set 100 samples. Spearman correlation was performed in R using the 'cor.test' function from the 'stats' package (R Core Team, 2019). We also visualised environmental drivers by plotting the distributions of cell concentrations and environmental parameters within the water column or within the first two axes of a Principal Component Analysis (PCA), and then interpolating values using the Multilevel B-spline Approximation (MBA) algorithm described by Lee et al. (1997). Prior to conducting the PCA, samples with a Cook's Distance greater than 4 times the sample size were removed. For the visualizations, we used the same environmental parameters and samples as for the Spearman correlations, except for the AMT data set where we plotted chlorophyll instead of depth - which allowed visualization of the deep chlorophyll maximum (DCM). For the AMT data set, we plotted the abundance and environmental parameters as a function of latitude and depth. While for the Mediterranean data set the variables were plotted as a function of the first two axes of a PCA which included temperature, salinity, day length, and concentrations of phosphate, DIN, and silicate. Two different strategies were used to visualize the AMT and Mediterranean data sets as the AMT data set is spatial and the Mediterranean data set is temporal. The MBA interpolation was performed with the 'mba.surf' function from the 'MBA' R package (Finley et al., 2017), and the PCA was performed with the 'dudi' function of the 'ade4' package (Dray and Dufour, 2007). Cook's distances were calculated using the 'lm' and 'cooks.distance' functions provided in the 'stats' R package (R Core Team, 2019).

## 2.4 Seasonality

To investigate seasonality we compared monthly hetero- and holococcolithophore abundance data to temporal variations of temperature, salinity, day length, and concentrations of phosphate, DIN, and silicate of the BATS and Mediterranean data sets.

## 2.5 Niche overlap and niche expansion

Distribution patterns of phytoplankton are influenced by multiple environmental drivers. These environmental drivers form a n-dimensional hyperspace within which hypervolumes can be defined based on where the phytoplankton occur. This hypervolume is considered to be the species niche space (Hutchinson, 1957) and allows niche comparisons between multiple phytoplankton - in this instance the two life cycle phases of coccolithophores.

Although processing hypervolumes is challenging due to their high dimensionality, methods described by Blonder et al. (2014) allow hypervolume quantification and comparison (for futher discussion see Blonder (2018) and Mammola (2019)).

Using this strategy we determine the niche overlap of hetero- and holococcolithophores in hyperspace using the Sørensen-Dice and Jaccard similarity metrics.

We furthermore calculate the 'niche expansion' of the haplo-diplontic life cycle strategy, which we define here as the non-overlapping region of either phase within hyperspace. In other words:

$$NE(A) = \frac{|A| - |A \cap B|}{|A \cup B|} \tag{2}$$

Where NE(A)= niche expansion of A; A = hypervolume A; B = hypervolume B; $\cap$ = intersection between two hypervolumes; $\cup$ = union between two hypervolumes

The niche metrics utilized in this study are illustrated in Fig. 3. Although we visualize the niche space of each species using contours, in reality the niche metrics are calculated based on random points sampled from the inferred hypervolumes (Blonder et al., 2014)

We calculated the Jaccard and Sørensen-Dice similarity metrics and niche expansion for the AMT, BATS and Mediterranean Sea data sets. For the AMT data set, DIN showed high Pearson correlation to silicate ($\rho = 0.95$, p < 0.001) and phosphate ($\rho = 0.90$, p < 0.001). We thus only considered temperature, salinity and the concentration of DIN in this region. Although no such correlations were observed for the Mediterranean data set, and weaker but significant relationships were observed in the BATS stations ($\rho = 0.74$, p < 0.001 for silicate and $\rho = 0.84$, p < 0.001 for phosphate), to make the niche metrics comparable in all regions, the silicate and phosphate concentration of the Mediterranean and BATS data set were also excluded.

It is likely however that silicate and phosphate as well as other parameters (such as irradiance, turbulence and carbonate chemistry) influence the niche space of coccolithophores and thus the metrics calculated. Beside the influence of environmental parameter choice, results of the niche space analysis will depend on what is considered a paired species. Although we use up to date definitions from Frada et al. (2018), these definitions are likely to change in the future. Finally, cryptic speciation (Geisen et al., 2002) and subsequently the pairing of multiple HOL phases to single HET phases and vice versa complicates results.

The environmental data were normalized using z-scores prior to analysis. Niche overlap and niche expansion was calculated only for species for which both life cycle phases were observed. In addition to calculating the niche expansion for individual species, we calculated a average niche expansion by taking the mean NE values of all individual species for both the haploid and diploid coccolithophores life cycle phases.

We used the 'hypervolume' R package (Blonder and David J. Harris, 2018) to conduct our niche overlap and niche expansion analysis. Gaussian kernel density estimation (R function 'hypervolume_gaussian') was used to construct the hypervolume, the overlap metrics were calculated with the 'hypervolume_overlap_statistics' R function, and the volume and intersection of hyper volumes were calculated using the 'get_volume' R function.

## 3 Results

### 3.1 Biogeography of coccolithophores

Within our compilation heterococcolithophores showed global distribution, while holococcolithophores were noticeably absent
at the ALOHA station in Hawaii and (with some exceptions) >50° S in the Southern Ocean (Fig. 4 and Table 3).

Highest maximum abundances of heterococcolithophores are observed at high latitudes within the Arctic circle (>66° N) ($\approx$4.37 x $10^6$ cells $1^{-1}$ for Jun-Aug), and the Southern Ocean (>40° S and <65° S) ($\approx$1.64 x $10^6$ cells $1^{-1}$ for Dec-Feb). Generally, maximum abundances above 1 x $10^5$ cells $1^{-1}$ were observed, except between Sep-Nov in the Indian Ocean ($\approx$3.33 x $10^4$ cells $1^{-1}$), Sep-Nov and Dec-Feb in the Atlantic Ocean ($\approx$5.40 x $10^4$ and $\approx$9.78 x $10^4$ cells $1^{-1}$ respectively) and Mar-May in the Pacific Ocean ($\approx$4.96x $10^4$ cells $1^{-1}$).

The regions and periods with the highest mean heterococcolithophore abundance differ from the regions/periods with the highest maximum heterococcolithophore abundance. For example, the highest mean abundance is observed in the Indian Ocean during Mar-May ($\approx$1.13 x $10^5$ ($\pm$ 2.97 x $10^4$) cells $1^{-1}$); which is higher than the highest mean abundance in the Southern Ocean observed during Mar-May ($\approx$1.17 x $10^5$ ($\pm$ 2.88 x $10^4$) cells $1^{-1}$), and in the Arctic Circle during Jun-Aug ($\approx$5.83 x $10^4$ ($\pm$ 2.97 x $10^4$) cells $1^{-1}$).

Although holococcolithophores show low abundances in the high latitudes of the Southern Hemisphere, highest maximum holococcolithophore abundances are observed in the Arctic circle (>66° N) during Jun-Aug ($\approx$2.23 x $10^5$ cells $1^{-1}$). High maximum abundances are additionally observed in the Mediterranean Sea (Sep-Nov) ($\approx$1.27 x $10^5$ cells $1^{-1}$).

The lowest maximum holococcolithophore abundance is observed in the Pacific Ocean during Jun-Aug (4.45 x $10^2$ cells $1^{-1}$) and in the Arctic Circle during Sep-Nov (1.12 x $10^3$ cells $1^{-1}$).

On average, the Mediterranean Sea has the highest mean holococcolithophore abundance (between $\approx$2.21 x $10^3$ and 9.42 x $10^3$ cells $1^{-1}$), followed by the Indian Ocean ($\approx$1.41 x $10^3$ - 4.80 x $10^3$ cells $1^{-1}$). The lowest mean abundances are observed in the Pacific Ocean (4.9 x $10^1$ ($\pm$ 9.70 x $10^1$), Arctic Circle (Sep-Nov; 2.55 x $10^2$ ($\pm$ 2.71 x $10^2$) and Southern Ocean (Dec-Feb; $\approx$3.24 x $10^2$ ($\pm$ 2.06 x $10^2$) cells $1^{-1}$).

Depending on the season holococcolithophore contribution to total coccolithophore abundance varies globally between 1.67 % ($\pm$ 0.37%) in Dec-Feb and 16.16 % ($\pm$ 1.68%) in Jun-Aug, with highest contribution observed in the Mediterranean Sea in Jun-Aug (31.38 % $\pm$ 2.93 %) (Table 3). On an regional scale outside of the Mediterranean Sea, holococcolithophores contribute less than 8 % to the total coccolithophore abundances. However, the contribution of holococcolithophores to paired species is higher than when all hetero- and holococcolithophores are considered (Table 4), with a HOLP-index between 5.65 % ($\pm$1.71 %) and 27.41 ($\pm$2.67%) globally depending on season. The lowest HOLP-indices were observed in the Atlantic Ocean (Sep-Nov) (0.59 $\pm$0.81) and Dec-Feb (0.47 $\pm$0.65%), and in the Southern Ocean (Dec-Feb) (0.61 $\pm$0.58%). The highest HOLP-index was observed in the Mediterranean Sea in between Jun-Aug (39.03 $\pm$3.23%).

## 3.2 Vertical distribution

In the global data set, heterococcolithophore abundance is evenly distributed with depth, while holococcolithophore abundance
is highest in the top 50 m of the water column (Fig. 5).

For holococcolithophores the vertical distribution pattern is mainly driven by paired holococcolithophore species which constituted ≈62.2 % to total coccolithophore abundance. Two currently unpaired holococcolithophores also contribute to the depth distribution trend with *Helladosphaera cornifera* (for which the association has to be further confirmed) constituting ≈8.1 % of total holococcolithophore abundance, and *Corisphaera gracilis* (for which no pair has been described) constituting ≈3.6 % of total holococcolithophore abundance. Subsequently paired holococcolithophore abundances broadly followed the same patterns observed when all holococcolithophores were considered.

In comparison to holococcolithophores, depth distribution of heterococcolithophores was driven by unpaired species - in particular *E. huxleyi* which constituted ≈59.2 % of total heterococcolithophore abundance, but also by the presence of unpaired deep water species such as *Ophiaster formosus*, *Florisphaera profunda*, *Calciopappus caudatus*, and *Oolithotus antillarum*. However, although paired heterococcolithophores only contributed ≈5.7 % to total heterococcolithophore abundance, the depth distribution trends of paired and total heterococcolithophores species were similar.

## 3.3 Environmental drivers of niche partitioning

To further understand the distribution patterns observed on a global basis and within the water column we investigated the environmental drivers of hetero- and holococcolithophore abundance in the Atlantic Ocean (with the AMT data set) and the Mediterranean Sea. For the Atlantic Ocean data set, the environmental drivers were considered in the context of their distribution within the water column, whereas for the Mediterranean the environmental drivers were considered within PCA 'niche space'. These observed patterns were then further corroborated through Spearman analysis.

### 3.3.1 Atlantic Ocean

In the Atlantic Ocean both hetero- and holococcolithophores have highest abundances in the top 50 m of the water column (Fig. 6). However, a noticeable difference between hetero- and holococcolithophore distribution (Fig. 6a and Fig. 6d respectively) is the absence of holococcolithophores below the deep chlorophyll maximum (DCM) (Fig. 6l). The DCM tends to occur at 1-10 % irradiance levels, and is closely linked to the nutricline and thermocline (Poulton et al., 2006). The difference in depth distribution between hetero- and holococcolithophores, and the absence of holococcolithophores below the DCM may therefore be influenced by a combination of light limitation, high nutrient concentrations, and cold water temperatures at depth or other factors not addressed in this study.

This suggests that heterococcolithophores might be better adapted to exploit such conditions. Although differences in sinking rates - which are conceivably higher in the more heavily calcified heterococcolithophores could also factor into the difference in depth distribution between the two life cycle phases.

The distribution of heterococcolithophores (Fig. 6a) is primarily driven by *E. huxleyi* (Fig. 6c) which constitutes ≈30 %
of total heterococcolithophore abundance in the data set. When only paired heterococcolithophore species were considered
(Fig. 6b) a more even distribution in subtropical and tropical regions is observed. Holococcolithophores and paired holococ-
colithophores showed roughly similar distribution patterns (Fig. 6d and Fig. 6e).

Within the upper water column, heterococcolithophores showed highest abundance at higher latitudes (>35° N and >30°
S), which is associated with a shallow mixed layer, lower salinity, and lower temperature, as well as increasing silicate concen-
trations in the southern hemisphere. Holococcolithophores meanwhile showed highest abundances at both high latitudes and
in the Atlantic subtropical gyres. The HOLP-index (Fig. 6f) was highest within the Atlantic subtropical gyres, with a higher
proportion of holococcolithophores in the Northern subtropical Gyre, which is associated with a shallower DCM relative to
the Southern subtropical Gyre. This shallowing of the DCM on the AMT is however likely a seasonal signal as described by
Poulton et al. (2006) and Poulton et al. (2017).

Spearman correlations (Table 6) suggests holococcolithophores are significantly ($p<0.05$) negatively correlated to phos-
phate, DIN, silicate and depth and significantly positively correlated to temperature and salinity. Paired holococcolithophore
and the HOLP-index showed the same correlation trends as holococcolithophores.

On the contrary, heterococcolithophores are only significantly and negatively correlated with depth and phosphate. While
for paired heterococcolithophores significant negative correlations were observed with depth and silicate.

Thus hetero- and holococcolithophore abundance in the Atlantic Ocean seems primarily driven by the depth of the DCM
both in terms of vertical and latitudinal distribution. Highest abundances of both hetero- and holococcolithophores are observed
above the DCM, and heterococcolithophores are present below the DCM while holococcolithophores are not. In terms of
latitude highest abundances of heterococcolithophores correspond to shallow DCM depth which occurs in higher latitude
regions, and highest abundances of holococcolithophores occur in subtropical regions with deep DCM depths.

### 3.3.2 Mediterranean Sea

For the Mediterranean Sea long term time series, niche separation of hetero- and holococcolithophores within the PCA niche
space (Fig. 7), is primarily driven by Principal Component 1 (PC1) which is positively associated with temperature and day
length and negatively associated with salinity, DIN, silicate and phosphate (see Table 7). Heterococcolithophores are most
abundant at low PC1 values (i.e. the left quadrants of Fig. 7a) which corresponds to low temperatures and short day lengths,
and high salinity and concentrations of DIN, silicate and phosphate. Holococcolithophores are most abundant at high PC1
values (i.e. the right quadrants of Fig. 7b), which corresponds to high temperatures and long day lengths, and low salinity and
concentrations of DIN, silicate and phosphate.

The pattern observed in the PCA niche space should be interpreted with some caution because only a portion of the variance
is captured (53%) and the use of interpolation introduces additional uncertainties. Besides, the structure of the PCA depends
highly on the number and type of variables included (Fig. S2), in particular when time is considered. However, the patterns
presented in the PCA are also apparent in the Spearman correlations (see Table 6), which suggests that the PCA plots are
qualitatively a good representation of the data.

The Spearman correlations indicate that heterococcolithophores are significantly negatively correlated to temperature, and day-length, and significantly positively correlated to phosphate, DIN, silicate and salinity. For paired heterococcolithophore species the only significant correlation observed was a positive correlation with silicate.

Holococcolithophores showed an opposite pattern to heterococcolithophores, and are significantly positively correlated to day-length and temperature, and significantly negatively correlated to salinity, DIN, silicate and phosphate. Paired holococcolithophores and the HOLP-index showed significant positive correlation to temperature and day length, but no significant correlations with the other environmental variables were observed.

### 3.3.3 General environmental trends

Our statistical analysis shows that in both the Mediterranean Sea and Atlantic Ocean holococcolithophores are generally found in low nutrient and warm environments and high light availability. However, an opposite trend was observed between the Atlantic Ocean and Mediterranean Sea in terms of correlation to salinity, with holococcolithophores positively correlated to salinity in the Atlantic Ocean and negatively correlated to salinity in the Mediterranean Sea. This difference in correlation to salinity may be explained by the different drivers of salinity in both regions. In the Atlantic Ocean, low salinity occurs at high latitudes, while high salinity corresponds to mid-ocean gyres due to higher evaporation in tropical and sub-tropical regions. In contrast, at the coastal site in the Mediterranean Sea, low salinity is strictly related to direct freshwater input and associated nutrients. As such salinity may be simply correlated to other environmental drivers, rather than be a driver itself.

In the Mediterranean Sea and the Atlantic Ocean significant negative correlations were observed between holococcolithophores and silicate. Although this correlation could be in part to strong correlation between DIN and Silicate ($\rho = 0.95$) observed in the Atlantic Ocean, the reason for this is less clear in the Mediterranean Sea as no such correlation is observed. A physiological reason for the negative correlation to silicate could be different silicate requirements among different coccolithophore species. Durak et al. (2016) for instance found evidence of silicate requirement for the heterococcolith life cycle phases of *S. apsteinii*, *C. coccolithus* and *C. leptoporus* but not for *E. huxleyi* or *G. oceanica*. Follow up experiments have furthermore found holococcolith life cycle phases of *C. coccolithus* and *C. leptoporus* do not require silicate (manuscript by Langer et al., in prep).

Statistically significant correlations were the same when all holococcolithophores, paired holococcolithophores or the HOLP-index was considered at both locations - however fewer significant correlations were observed for paired holococcolithophores and the HOLP-index.

The trend for heterococcolithophores is less clear when comparing the two sites: an opposite trend to holococcolithophores - e.g. high nutrients and low temperatures - is observed in the Mediterranean Sea, but not in the Atlantic Ocean where many of the correlations were not significant and heterococcolithophore were negatively correlated to phosphate. This negative correlation to phosphate is potentially due to deeper sampling in the Atlantic Ocean combined with high phosphate concentrations in deep and light limited waters skewing correlations, which highlights the need to consider sampling and DCM depth when comparing environmental correlation between studies. It may furthermore be due to the presence of mixotrophic or heterotrophic coccolithophores at depth in the Atlantic Ocean, which are not found in the shallow coastal waters of the Mediterranean Sea.

### 3.4 Niche overlap and niche expansion

We conducted niche similarity and niche expansion calculations on both the AMT, BATS and Mediterranean data sets to quantify niche space in these regions. For niche overlap we considered the Jaccard overlap and Sørensen-Dice overlap metrics which range from 0 to 1, with 1 signifying complete overlap. For niche expansion we considered the relative amount each life cycle contributed to the total niche space. In the AMT the niche overlap of paired species was high for both the Jaccard overlap and Sørensen-Dice overlap metrics (0.84 and 0.91 respectively, Table 8). However, for individual species the overlap metrics were highly variable ranging from 0.11 - 0.74 and from 0.20 - 0.81 for the Jaccard overlap and Sørensen-Dice overlap metrics, respectively. The niche expansion was higher for heterococcolithophores than holococcolithophores when all paired species were considered (see Table 8), but was again highly variable for individual species. The holococcolithophore phase of *C. mediterranea*, *S. bannockii*, *H. wallichii*, and *C. leptoporus* for instance all contributed more to the total niche space than their heterococcolithophore life cycle phase.

For BATS, the niche overlap values are generally smaller than for the AMT, with a Jaccard overlap and Sørensen-Dice values of 0.60 and 0.66, respectively. The niche expansion of heterococcolithophores at BATS is larger compared to the AMT (0.49 versus 0.11 for BATS and AMT, respectively). The NE of holococcolithophores is similar for both stations (0.02 versus 0.05 for BATS and the AMT, respectively). *S. anthos* and *S. pulchra* are the only species for which coccolithophore life cycle pairs are observed at the BATS station. For these species, the NE of heterococcolithophore is similar to when all species were considered, but is higher for holococcolithophores. In the Mediterranean Sea, the niche overlap and niche expansion values are more similar to the BATS data set than to the AMT data set.

Niche expansion of heterococcolithophores was also higher than holococcolithophores when all paired species were considered, but like in the Atlantic Ocean species specific exceptions were observed. The holococcolithophore phase of *C. mediterranea*, *S. histrica*, *S. strigilis* and *C. leptoporus* all contributed more to the total niche space than their heterococcolithophore life cycle phase in this region. In the Mediterranean Sea the niche space of *S. molischii* is of particular note, as no overlap between the two life cycle phases was observed, and the two unique components were of similar size (0.51 and 0.49 for hetero- and holococcolithophores respectively).

Although quantitative interpretation of niche space is difficult since niche space will vary depending on the number of environmental axes included (Blonder et al., 2014), these results highlight that holococcolithophores contribute significantly to the niche space of coccolithophores, in some instances contributing more to total niche space than the heterococcolithophore phase. In this context *C. pelagicus* is particularly relevant as this species contributes significantly to the global carbonate flux (Ziveri et al. (2007); Rigual Hernández et al. (2019), and is one of the key calcifiers in the Arctic Ocean (Daniels et al., 2016).

These results additionally suggest that the niche expansion patterns of the coccolithophore life cycle are more similar between the BATS and Mediterranean Sea than BATS and the AMT. This suggest that seasonal variations play an important role in structuring the niche space of coccolithophores, otherwise BATS and the AMT should be more alike due to more similar hydro-graphic conditions. This result highlights the value of time series for studying the ecology of the coccolithophore life cycle and raises the need for caution when comparing niche space of cruise data and time series.

## 3.5 Seasonality of coccolithophores

Hetero- and holococcolithophore abundance highly varies with season at both the BATS station in the Atlantic Ocean and the long-term stations in the Mediterranean sea (Fig. 8). Both locations experience a peak of heterococcolithophores in the winter, followed by a peak of holococcolithophores at the end of spring, and in early summer. In the Atlantic Ocean, the heterococcolithophore are present in high abundance for a longer period of time - overlapping with the spring peak in holococcolithophore abundance (Fig. 8b).

At both locations the peak of the holococcolithophore bloom occurs in the spring and summer when water temperatures rise and the day length is longest, while heterococcolithophore abundance is highest in the winter when temperature is lowest and day length shortest. The seasonality of peak hetero- and holococcolithophore abundance may furthermore correspond to seasonal changes in mixed layer depth (MLD), as both the Atlantic Ocean and Mediterranean Sea experience increased mixing in the winter and higher stratification in the summer.

Clear seasonal patterns are observed for DIN and silicate concentrations at both locations. High holococcolithophore abundances are observed when silicate and DIN concentrations are low, and vice versa for high heterococcolithophore abundances. This observed seasonal patterns at the BATS and Mediterranean time series are thus consistent with our Spearman correlations (see Table 6 and Table 7 and our discussion above) and PCA niche space (see Fig. 7 and discussion above).

It is important to note that on a species level, individual species do not exclusively follow the seasonal hetero-holococcolithophore trends described above, as illustrated in detail by the original publications (Cerino et al., 2017; Godrijan et al., 2018). For instance for *Syracosphaera molischii* and *Syracosphaera pulchra* the holococcolith rather than heterococcolith phase is the dominant life cycle phase in these time series. Furthermore the holococcolithophore phases of *S. molischii*, *Syracosphaera histrica*, *Algirosphaera robusta* and *Acanthoica quattrospina* are observed in the winter - a period when total holococcolithophore abundance is lowest. Finally on a individual level, succession does not immediately follow the previous life cycle phase with several months of absence observed between peak abundance for some species (Cerino et al., 2017; Godrijan et al., 2018).

This highlights that grouped hetero-holococcolithophore abundances represents a generalization that might not always represent patterns observed for individual species. These differences from generally observed patterns could be due to variations in life strategy - such as mixotrophy, motility and grazing susceptibility - independent of life cycle phase. Suggesting that functional traits different from the life cycle phase may determine the niche space these species inhabit.

## 4 Discussion

Our meta-analysis shows that holococcolithophores are a minor contributor to coccolithophore abundance in the modern ocean, contributing between $\approx$2-15 % to the total coccolithophore abundance and between $\approx$5-30% of the total paired coccolithophore abundance depending on season. However, our analysis also shows that haploid cells play an important role in coccolithophore ecology, accounting for $\approx 19\%$ of their niche space, which lesser or greater contributions depending on the species (3-76%). Our analysis furthermore shows that if conditions are favorable (specifically increased stratification and reduced nutrient supply) holococcolithophores can be significant contributors to the coccolithophore standing stock (up to $\approx 30\%$).

Although holococcolithophore contribution to calcium carbonate production is likely small due to their lower cellular $CaCO_3$ content - which is an order of magnitude lower than heterococcolithophores (Daniels et al., 2016; Fiorini et al., 2011a, b) - their role in the carbonate cycle in present, past and future oceans could have other biogeochemical effects. A shift towards a higher proportion of holococcolithophore cells, would result in lower global calcium carbonate production which could subsequently result in lower $CO_2$ outgassing on short time scales. Furthermore the ballasting effect of coccolithophores would be reduced if a shift towards more lightly calcified haploid cells occurred (Hoffmann et al., 2015) which would potentially reduce efficiency of the carbon pump by reducing sinking rates. Although how other factors such as shifts in carbonate chemistry impact holococcolithophore abundance are not clear, increased stratification and decreased nutrient supply are projected under the RCP 8.5 climate change scenario (Fu et al., 2016), which would favor holococcolithophores. This shift from diploid to haploid coccolithophores could on the one hand reduce $CO_2$ outgassing, but would additionally reduce ballasting and subsequently impact the carbon pump by reducing sinking rates.

In terms of the ecological niche space - which is the environmental range a species inhabits - observations of hetero- and holo-coccolithophores in our meta-analysis broadly conform to the Margalef Niche Space Model. This model was proposed by Margalef (1978), and posits that the distribution of phytoplankton functional groups relate broadly to turbulence, light and nutrients. Although we do not explicitly represent the former, turbulence is implicitly represented in our analysis based on mixed-layer depth. Within the Margalef Niche Space framework, we find that hetero- and holo-coccolithophores occupy an intermediate functional group located between diatoms and dinoflagellates (see Fig. 9) as proposed by Houdan et al. (2006) and Frada et al. (2018).

Diploid heterococcolithophores thus favour high nutrient, and more turbulent waters, whereas haploid holococcolithophores favour low nutrient, and more stratified waters.

Although the Margalef niche space model certainly presents a simplification that is prone to exceptions both for diatoms (see Kemp and Villareal (2018)) as well as coccolithophores (The generalist *E. huxleyi* and deep water species such as *F. profunda* are clear examples), the model broadly holds in our meta-analysis.

This ecological-environmental distinction of hetero- and holococcolithophores is observed in coccolithophore species distribution in our analysis in terms of geographical succession, depth distribution and seasonal trends. In the Southern Ocean and Atlantic Ocean a geographical shift from holococcolithophores to heterococcolithophores is observed as latitude and turbulence and nutrients increases. While in the Atlantic Ocean as well as the global data set a vertical shift is observed with holococcolithophores absent or at low abundance in deep nutrient rich waters. Finally, in the Mediterranean Sea, a seasonal shift is observed as heterococcolithophores are most abundant in well mixed nutrient rich winter months and holococcolithophore are most abundant in nutrient poor stratified summer months.

However, some exceptions occur. For instance in the AMT data set, although heterococcolithophores are more evenly distributed with depth, maximum abundance of heterococcolithophores is in surface waters, and subsequently heterococcolithophores are negatively correlated to nutrients. Nonetheless the relation to turbulence holds: heterococcolithophore abundance is highest in well mixed high latitude waters and holococcolithophore abundance is highest in stratified sub-tropical regions. Finally, many species specific exceptions occur. We highlight examples on a seasonal scale in our Mediterranean data set dis-

cussion (see Sect. 3.6), but exceptions were also noted along the AMT (see discussion in Poulton et al. (2017), and in other Mediterranean studies (Šupraha et al., 2016; D'Amario et al., 2017; Skejić et al., 2018). Which means that caution should be used when considering the niche space model for individual species.

## 4.1 Niche overlap and expansion

Our study showed that the niche volume of coccolithophores is larger when holococcolithophores are included in coccolithophore niche space. This tells us two things: first, studies focused solely on heterococcolithophore are underestimating coccolithophore habitat and thus inaccurately represent the coccolithophore functional group in modelling and physiological studies, which means that we might be underestimating their ability to compete with other phytoplankton, as well as the range of environmental conditions they can tolerate. Secondly, we underestimate coccolithophore primary productivity and calcite standing stock by not including accurate assessments of their abundance.

This might be of particular relevance for *E. huxleyi*, the diploid phase of which has been of particular research focus due to high abundances (approx 59.2% in our compilation). Although our meta-analysis does not include haploid abundance data of this species, we suspect, following upon our findings on the haploid/diploid paired species, that the haploid phase of *E. huxleyi* is also ecologically relevant. Previous studies suggest that the haploid life cycle phase of *E. huxleyi* can increase its niche space due to streamlined metabolism (Rokitta et al., 2011), and variations in response to bacterial (Mayers et al., 2016; Bramucci et al., 2018), and viral pressures (Frada et al., 2008). Although it should be noted that in some instance, morphology rather than ploidy level seems to be the primary driver for observed differences in *E. huxleyi* (Frada et al., 2017). Overall, observations in the haploid stage of *E. huxleyi* are extremely limited due to difficulty of identifying the haploid phase with regular light microscopy, highlighting the need for developing new techniques to account for this potentially important life cycle stage. Further development of FISH (Campbell et al., 1994) and COD-FISH (**?**) would be particularly relevant in this context.

## 4.2 Concluding remarks

Our compilation provides insight into the distribution of hetero- and holococcolithophores, but also highlights many gaps in the data distribution and our knowledge on coccolithophore ecology.

There is for instance a lack of SEM observations in the Pacific Ocean (2 studies in this compilation). Although, this is in part because existing data from the 1980s was not retrievable, and because the low abundance of coccolithophores in this region means that it has been of low priority for time-intensive and costly re-sampling. In addition, there is a limited number of available SEM time series, which are particularly valuable due to the seasonal nature of these organisms and the importance of time in structuring coccolithophore niche space. Patchiness of data combined with the patchiness of coccolithophore blooms is a challenge in fully assessing marine ecosystem functioning, and in providing global abundance estimations.

Nonetheless, from our compilation it is clear that holococcolithophores constitute a minor component of total coccolithophore abundance. This could be in part to sampling bias, specifically temporal bias to periods of high heterococcolithophore abundance. However, other factors such as the strong dominance of *E. huxleyi* which has a naked haploid phase, and the limited biomass low nutrient regions are able to sustain might also exert a significant influence. The low contribution of holo-

coccolithophores is interesting and raises the question what physiological traits make heterococcolithophores generally more successful in the modern ocean.

Beside limitations of in situ measurements, size, POC and PIC measurements of paired hetero- and holococcolithophore species are sparse, in particularly for holococcolithophores.

Such measurements are needed for global organic carbon and carbonate production estimates, which are critical for biogeochemical estimates, including model studies. Models which could then be used to contextualize in situ observations in biogeochemical context, and which could test response to environmental pressures presented by anthropogenic $CO_2$ emissions. Modelling approaches could furthermore be used to investigate drivers of distribution trends difficult to acquire with in situ measurements such as the role of competition with other phytoplankton and the influence of top down control on distribution trends, both of which have shown to be important drivers of coccolithophore distribution in previous studies (Monteiro et al., 2016; Nissen et al., 2018).

A pertinent environmental driver not covered in our meta-analysis due to limited data, is the influence of carbonate chemistry within the haploid-diploid niche space. As the haploid and diploid phases of coccolithophores vary in their calcification status, they may thus show different responses to carbonate chemistry. A study by Triantaphyllou et al. (2018) for instance found that holococcolithophores increased abundance in low pH waters. If this holds true on a global level, and holococcolithophores inhabit lower pH waters in terms of their niche space, this would have important implication in the context of ocean acidification. In particular because meta-analysis (Ridgwell et al., 2009; Krumhardt et al., 2017) and modelling (Ridgwell et al., 2007; Krumhardt et al., 2019) suggest a shift towards lower global calcification rates in response to ocean acidification and warming. It should however be noted that the response of heterococcolithophores to ocean acidification is both strain and species dependent (Langer et al., 2006, 2009; Meyer and Riebesell, 2015), and global calcification rates might be more impacted by shifts in species composition rather than individual response (Ridgwell et al., 2009). Furthermore, contradicting evidence suggesting increased coccolithophore abundance in response to higher $CO_2$ has been noted in situ (Rivero-Calle et al., 2015)

Finally, additional experiments on the numerical response of hetero- and holococcolithophores to various environmental drivers such as those performed on *E. huxleyi* would allow a better understanding of individual environmental pressures, and will furthermore be highly valuable for future modelling approaches. In this context a better understanding of the triggers of phase transition would additionally be highly desirable, as the lack of haploid-diploid pairs of the same strain limits genomic approaches.

## 5  Conclusions

Our analysis shows that holococcolithophores constitute a minor proportion of total coccolithophore abundance $\approx$2-15 %), and constitute about $\approx$5-30% of total paired coccolithophore abundance depending on season.

Our study furthermore shows that hetero- and holococcolithophores have contrasting environmental preference, and that therefore the haplo-diplontic life cycle expands the niche space coccolithophores can inhabit by $\approx$17 %. Although our findings are limited to holococcolith forming species, lab studies suggest similar patterns are likely to be observed for other coccol-

510 ithophore species such as *E. huxleyi*, and raises the question how much the haploid phase of this species contributes to global coccolithophore abundance.

These results highlight the need to include haploid cells into coccolithophore studies, both in the context of environmental studies, modelling approaches, and physiological studies. We limit our understanding of these organisms by just focusing on one life cycle phase, particularly in the context of coccolithophore response to climate change, as increased stratification in a
515 warming climate may favour the haploid life cycle of coccolithophores.

*Data availability.* The data from the compilation will be available on PANGAEA

*Author contributions.* JV, FM, GW and CB conceptualized the manuscript. AP, JG, and FC provided data for analysis. JV curated the data, performed the formal analysis and visualized the results. JV, FM, CB, AP, JG, FC, GL, and EM interpreted the results. JV and FM prepared the manuscript with contributions from all co-authors.

*Competing interests.* The authors declare that they have no conflict of interest

*Acknowledgements.* This research was supported by a NERC GW4+ DTP and the Natural Environment Research Council (NERC) [NE/L002434/1] studentship to JV. It was also supported by the Natural Environment Research Council grant (NE/N011708/1) to CB, GW, FM, and FM; the European Research Council grant ERC-ADG-670390 to CB; the NERC consortium grant (NER/O/S/2001/00680) and a NERC Fellowship (NE/F015054/1) to AP; and the Croatian Ministry of Science, Education and Sports (No. 098-0982705-2731) and European Community
Research Infrastructure Action under the FP7 "Capacities" Program (SYNTHESYS, Project GB-TAF-132) to JG. The Atlantic Meridional Transect is funded by the UK Natural Environment Research Council through its National Capability Long-term Single Centre Science Programme, Climate Linked Atlantic Sector Science (grant number NE/R015953/1). This study contributes to the international IMBeR project and is contribution number 349 of the AMT programme. The C1-LTER station is part of the national and international Long Term Ecological Research network (LTER-Italy, LTER-Europe, ILTER). Data for the C1-LTER station were obtained in the framework of the EU FP7
MedSeA (Mediterranean Sea Acidification in a Changing Climate) project. Environmental and nutrient data were made available through the OGS Italian National Oceanographic Data Center (NODC). This is a scientific contribution of Project MIUR - Dipartimenti di Eccellenza 2018-2022 to the DISAT of Milano-Bicocca. We would like to thank for the statistical clinics provided by the Institute for Statistical Science at the University of Bristol. Finally we would like to thank everyone who as contributed data to the compilation.

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

## A. Haplontic life cycle (Dinoflagellates)

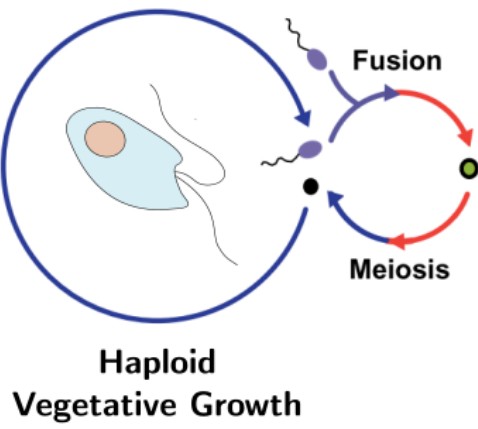

## B. Diplontic life cycle (Diatoms)

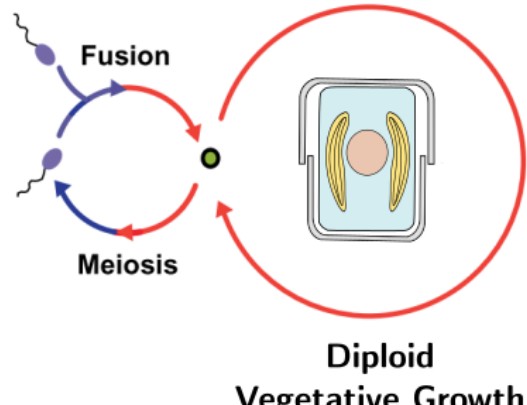

## C. Haplo-diplontic life cycle (Coccolithophores)

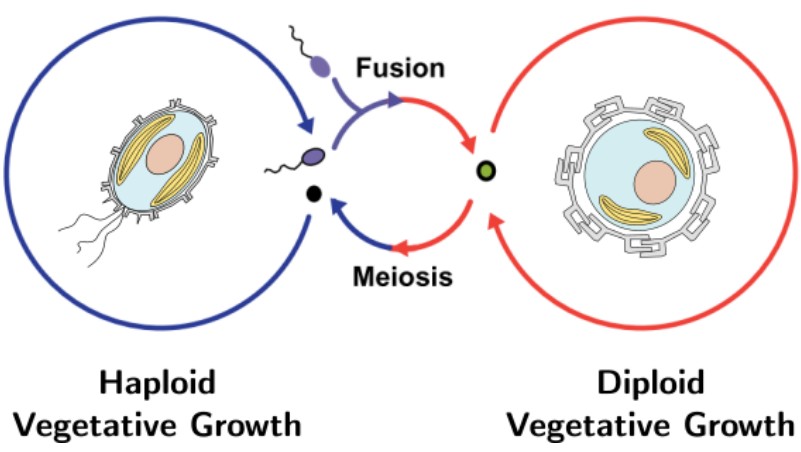

**Figure 1.** Life cycle strategies of phytoplankton. **(a)** Dinoflagellates tend to utilize a haplontic life cycle; **(b)** Diatoms tend to utilize a diplontic life cycle; **(c)** coccolithophores tend to utilize a haplo-diplontic life cycle. Note that not all coccolithophores calcify in their haploid phase.

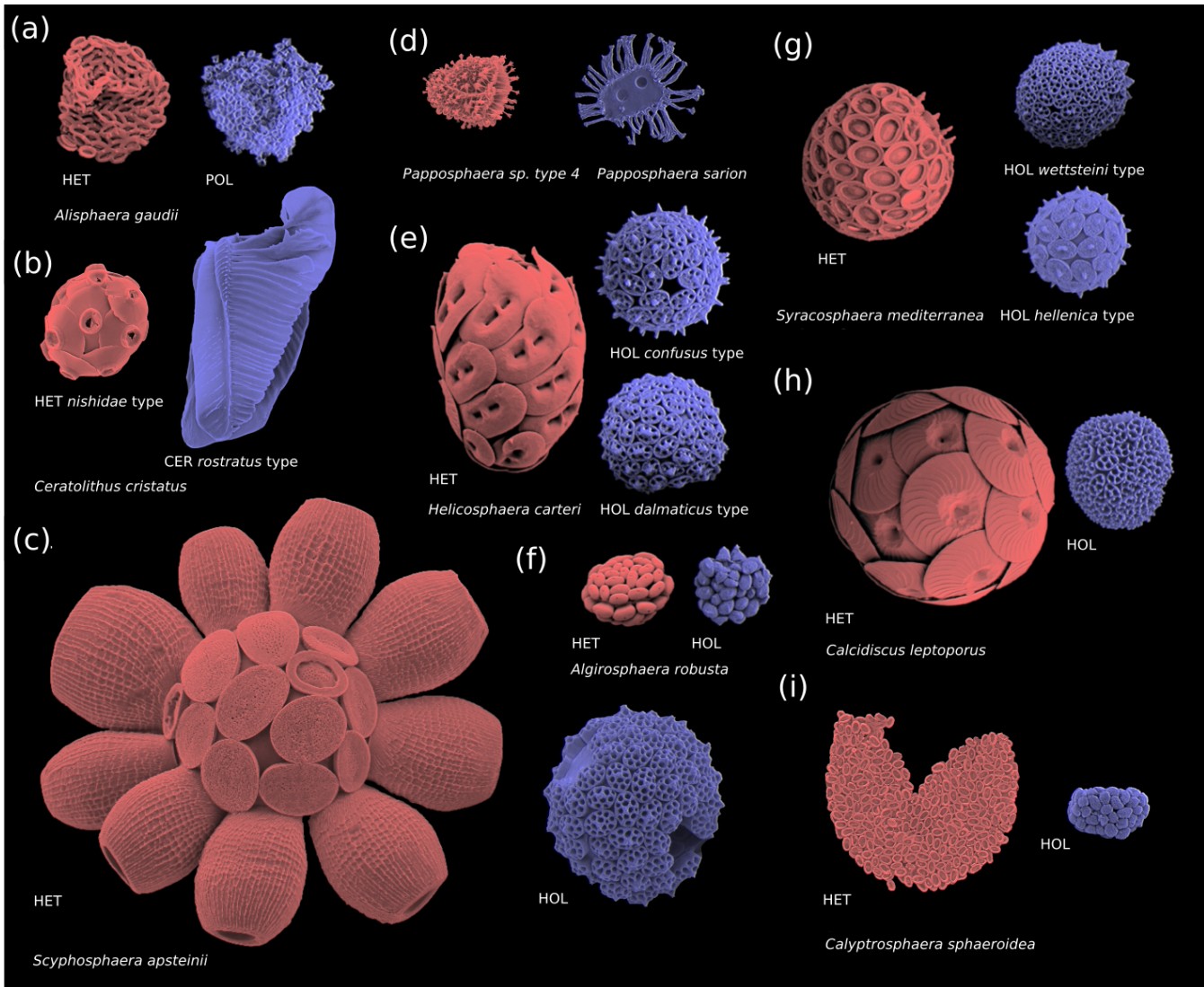

**Figure 2.** Coccosphere diversity of common cococcolithophores. Haploid cells are colored in blue and diploid cells in red. **(a)** Polycrater haploid morphology; **(b)** Ceratolith haploid morphology; **(c-i)**. Holococcolith haploid morphology. Note that in some instances multiple haploid phases are associated with one diploid phase (e.g. *S. mediterranea* and *H. carteri*), which may be due to cryptic speciation (Geisen et al., 2002). Furthermore, some species (e.g. *E. huxleyi*) do not calcify in their haploid phase and are thus not pictured. Images reproduced with permission from Young et al. **(b-d, i)**, and Šupraha et al. (2016) **(a, e-h)**. Images **(b-d, i (HOL))** by Jeremy Young, **(i (HET))** by Marie-Helene Kawachi, **(a, e-h)** by Luka Šupraha.

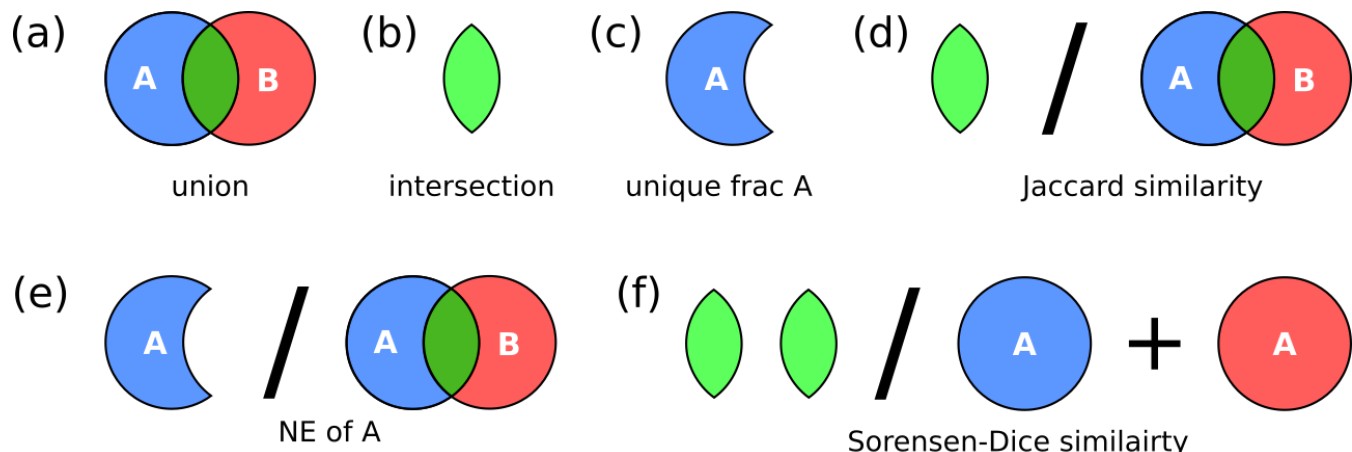

**Figure 3.** Hyper volume metrics utilized in this study. **a)** union; **b)** intersection; **c)** unique fraction A; **d)** Jaccard similarity metric; **e)** niche expansion: **f)** Sorensen-Dice similarity

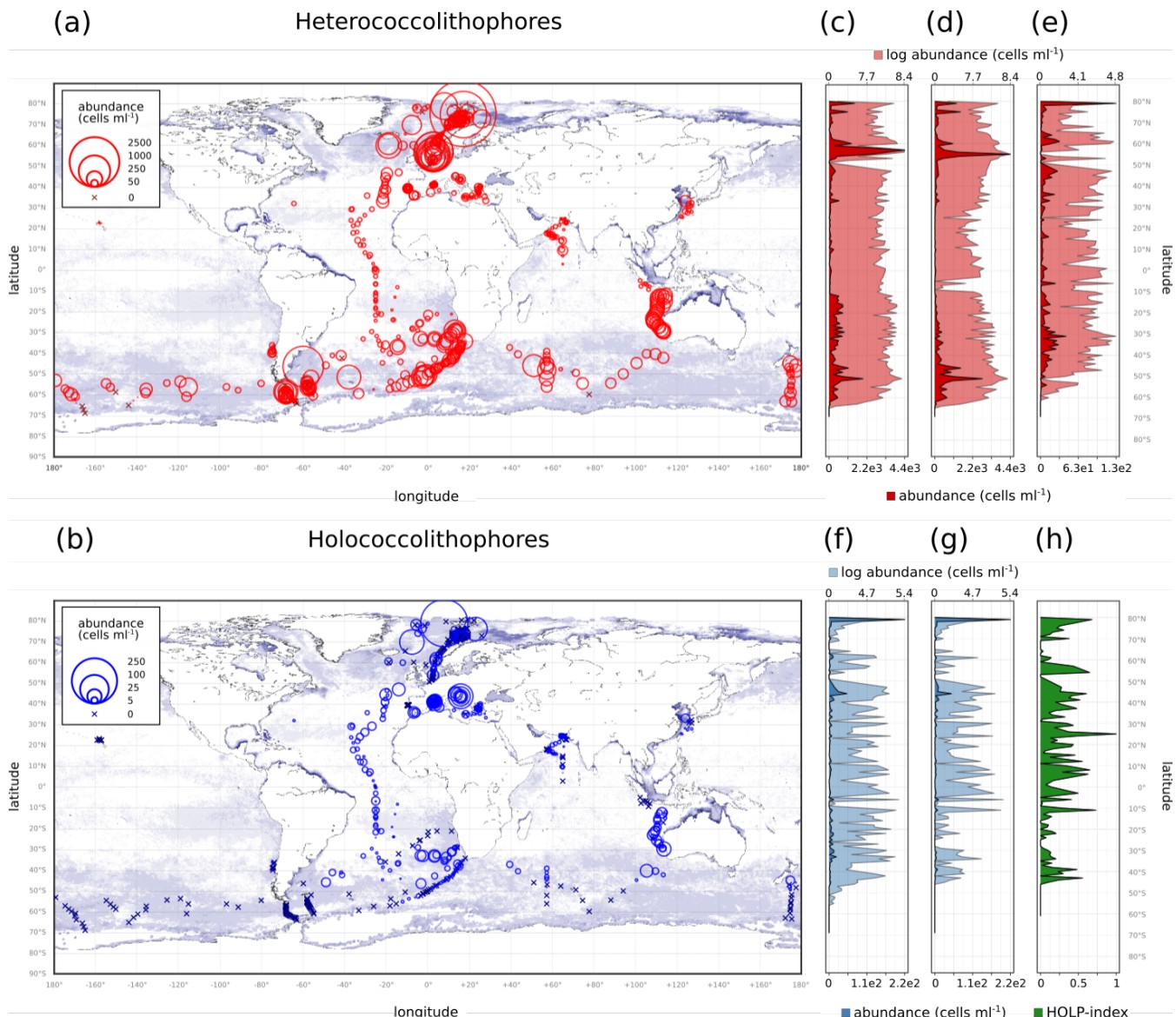

**Figure 4.** **(a-b)** Global coccolithophore distribution; **(c-h)** latitudinal coccolithophore distribution. **(a)** Heterococolithophores; **(b)** Holococolithophores; **(c)** Heterococcolithophores; **(d)** *E. huxleyi*; **(e)** Paired heterococolithophores; **(f)** Holococcolithophores; **(g)** Paired holococolithophores; **(h)** HOLP-index. For the latitudinal plots, the light shading is log transformed distribution.

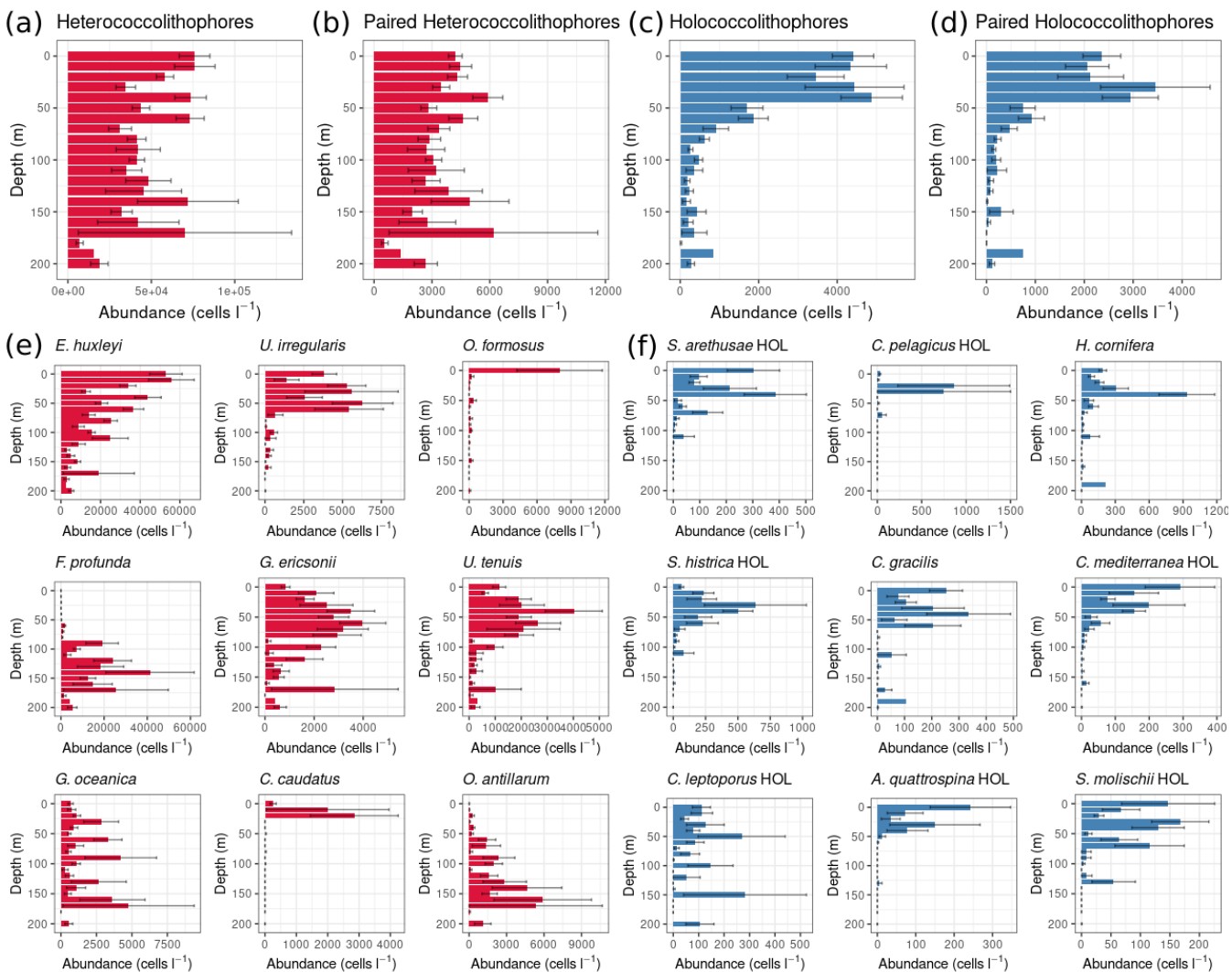

**Figure 5.** Global depth distribution of hetero- and holococcolithophores. **(a-d)** total paired and unpaired hetero- and holococcolithophore abundance; **(e-f)** individual species abundances. Heterococcolithophores are plotted in red, and holococcolithophores are plotted in blue. Only the most abundant coccolithophore species are plotted individually. Error bars are standard error.

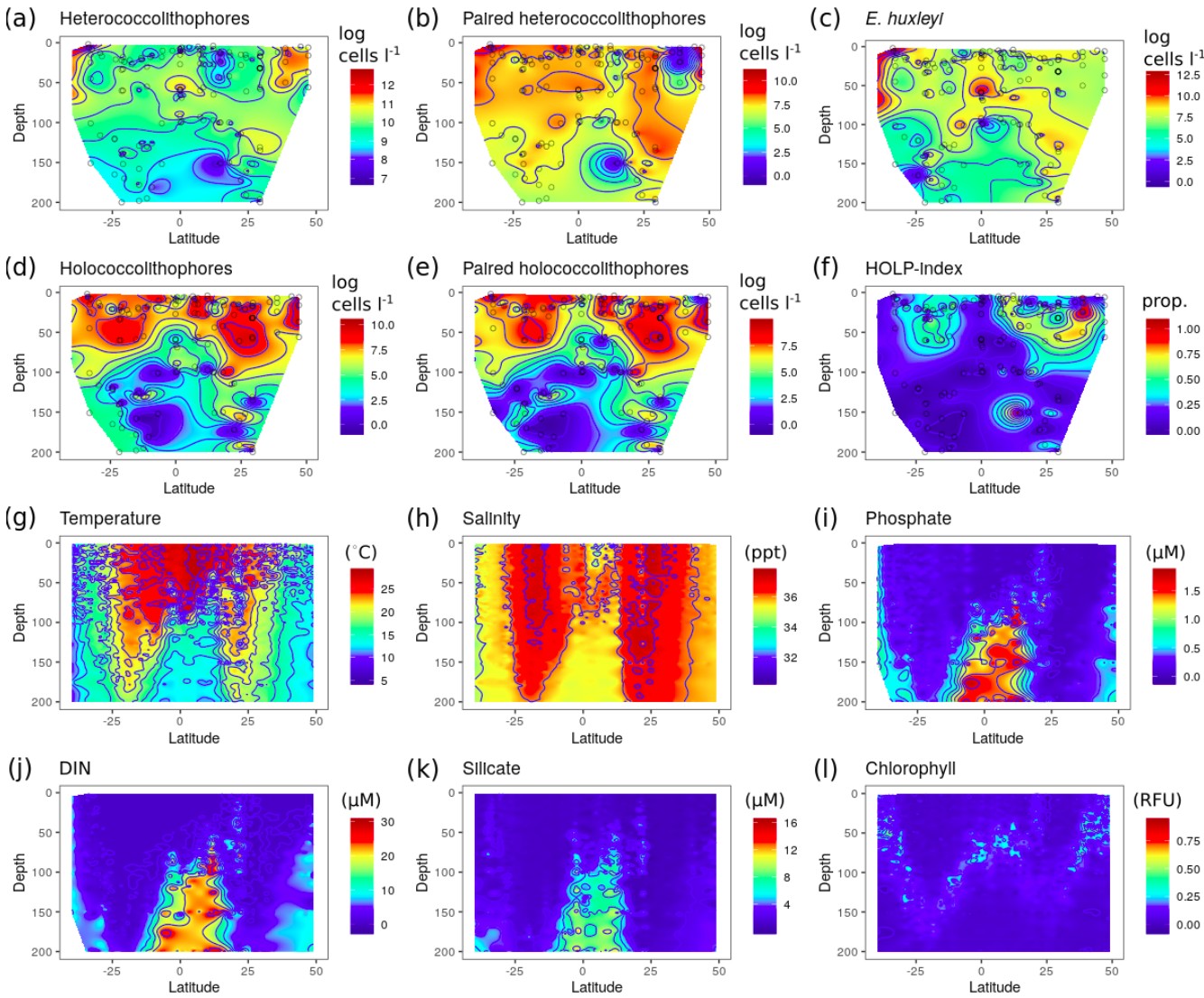

**Figure 6.** Depth distribution along AMT. **(a)** Heterococcolithophore abundance; **(b)** Paired heterococcolithophore abundance; **(c)** *E. huxleyi* abundance; **(d)** Holococcolithophore abundance; **(e)** Paired holococcolithophore abundance; **(f)** HOLP-index; **(g)** Temperature (° C); **(h)** Salinity (ppt); **(i)** Phosphate ($\mu$M); **(j)** Silicate ($\mu$M); **(l)** Chlorophyll. Species abundances are plotted on log scale.

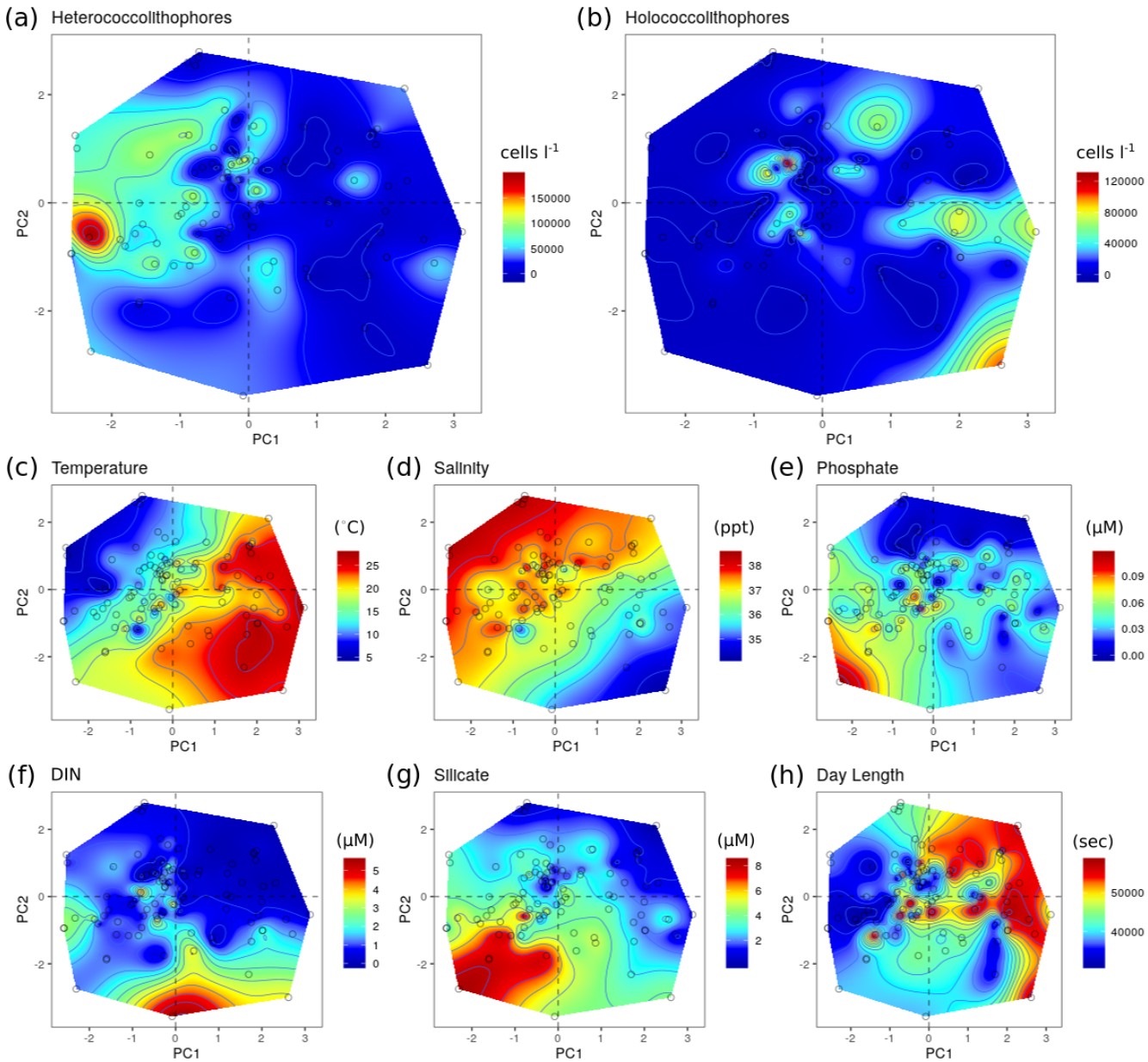

**Figure 7.** Principal Component analysis (PCA) of the RV-001 and LTER1 stations in the Mediterranean Sea. Abundance and environmental values were projected on the PCA *post hoc*, and then interpolated. **(a)** Heterococcolithophore abundance; **(b)** Holococcolithophore abundance; **(c)** Salinity; **(d)** Temperature; **(e)** Depth; **(f)** Phosphate; **(g)** DIN; **(h)** Silicate. Data was acquired from Cerino et al. (2017) and Godrijan et al. (2018).

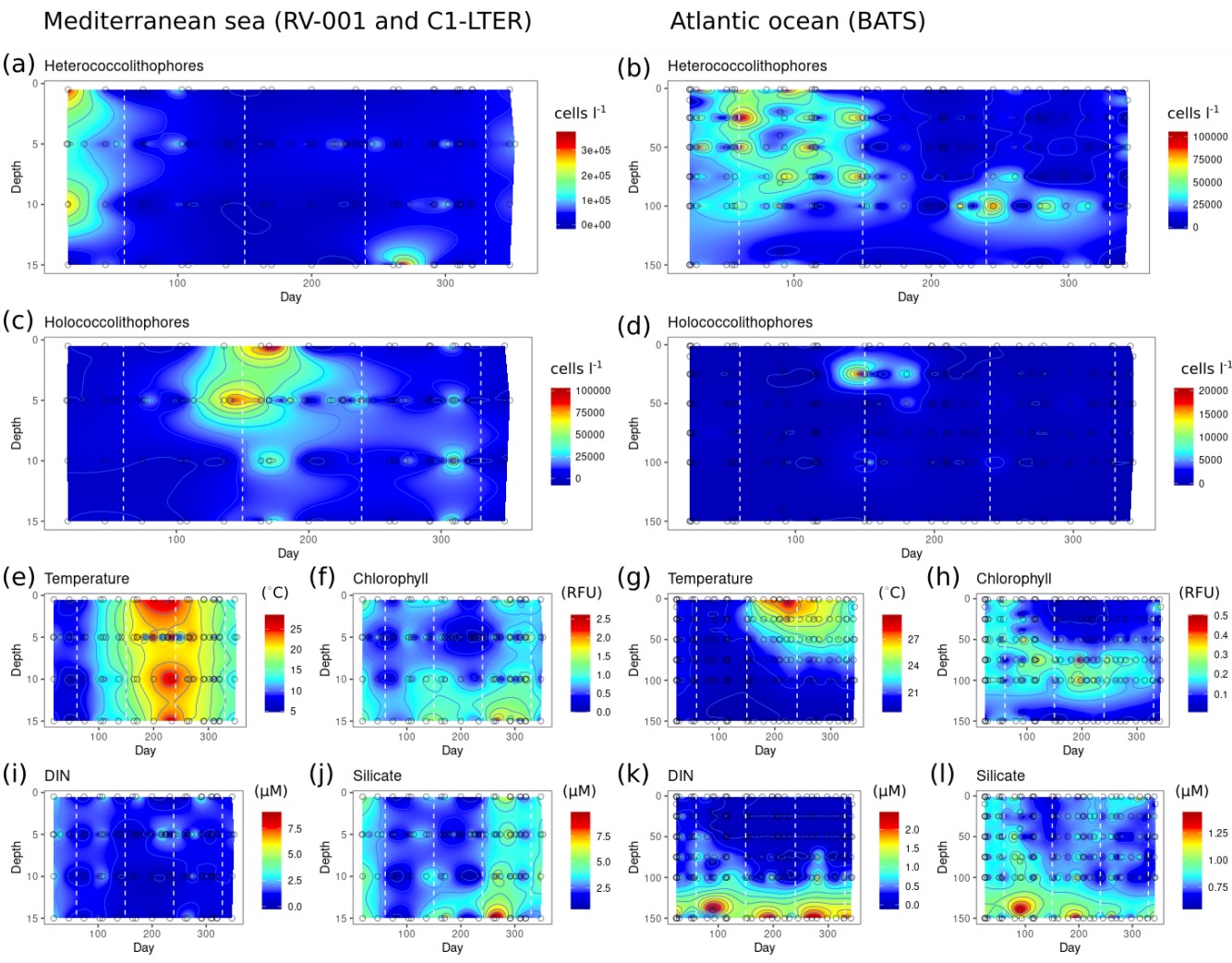

**Figure 8.** Seasonality of hetero- and holococcolithophores at the BATS station in Bermuda (left column) and the RV-001 and LTER-1 stations in the Mediterranean Sea (right column). Note heterococcolithophores are most abundant in the winter followed by a high abundance of holococcolithophores in the late spring and early summer. **(a-b)** Heterococcolithophore abundance; **(b-c)** Holococcolithophore abundance; **(e, g)** Temperature; **(f, h)** Chlorophyll; **(i, k)** DIN (nitrite + nitrate); **(j, l)** Silicate.

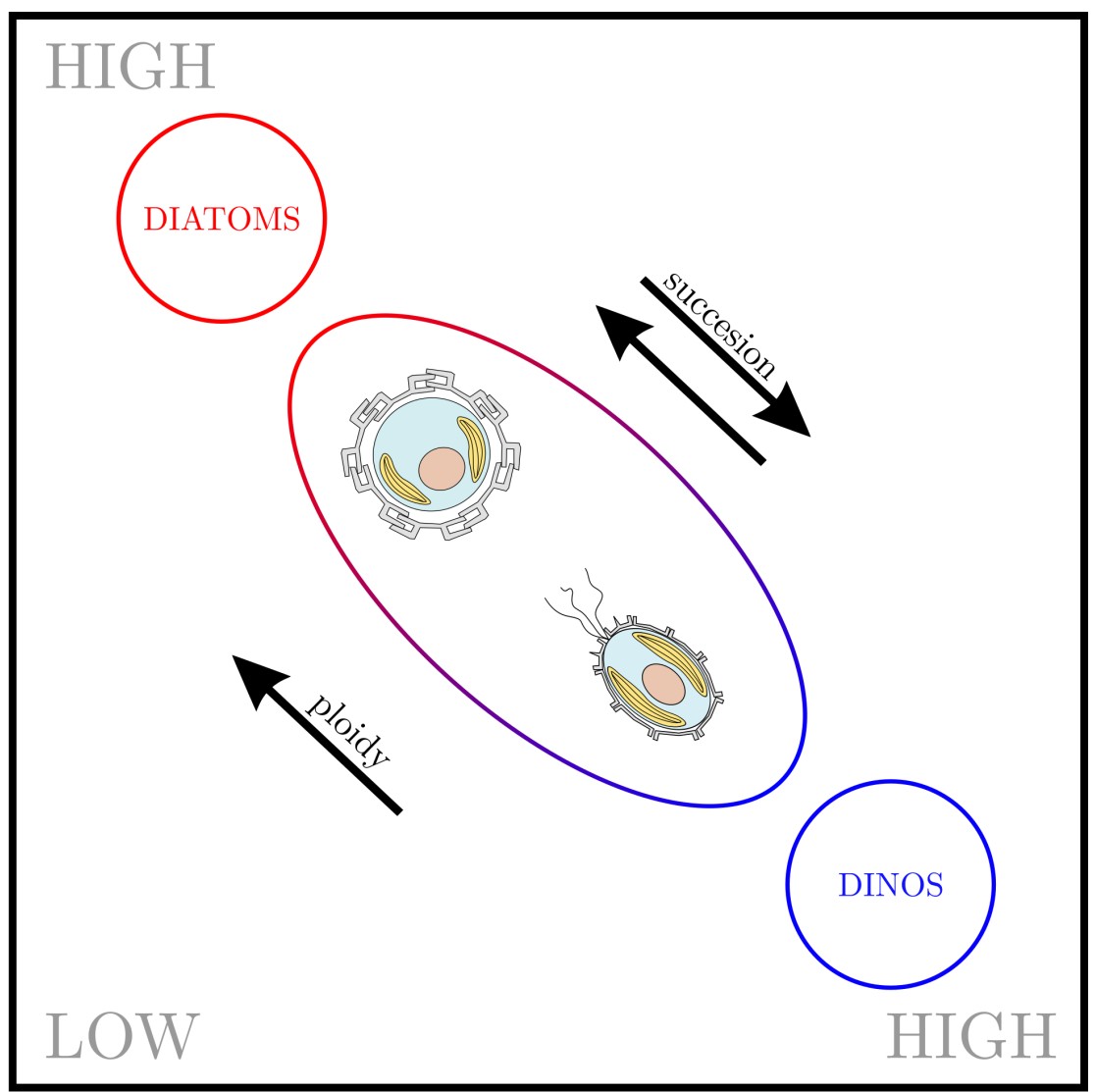

**Figure 9.** A. Modified version of Margalef's niche space model (Margalef, 1978) as proposed by Houdan et al. (2006) and Frada et al. (2018). Note that we have added day length, which was proposed by Balch (2004) as a third axis of the Margalef niche space model.

**Table 1.** Overview of Metadata.

| Reference | Survey Period | Region | Method | HOLP | n |
|---|---|---|---|---|---|
| Andruleit et al. (2003) | Sep (1993) | Arabian Sea | SEM | YES | 71 |
| Andruleit (2005) | Jun (2000) | Arabian Sea | SEM | NO | 21 |
| Andruleit (2007) | Jan to Feb (1999) | Indian Ocean | SEM | YES | 45 |
| Boeckel and Baumann (2008) | Mar to May (1998) | South Atlantic | SEM | YES | 57 |
| | Feb to Mar (2000) | | | | |
| Baumann et al. (2008) | Feb (1993, 1996) | South Atlantic | SEM | NO | 34 |
| | Mar (1996), Dec (1999) | | | | |
| Cerino et al. (2017) | Monthly (2011-2013) | Mediterranean sea | pLM/SEM | YES | 84 |
| Charalampopoulou et al. (2011) | Jul to Aug (2008) | North Sea and Arctic Ocean | SEM | YES | 94 |
| Charalampopoulou et al. (2016) | Feb to Mar (2009) | Southern Ocean | SEM | YES | 103 |
| Cepek (1996) | Feb (1993) | South Atlantic Ocean | SEM | YES | 33 |
| Cros and Estrada (2013) | Jun to Jul, and Sep (1996) | Mediterranean sea | SEM | YES | 113 |
| ? | Apr (2011) and May (2013) | Mediterranean sea | SEM | YES | 44 |
| Daniels et al. (2016) | Jun (2012) | Arctic Ocean | pLM/SEM | YES | 19 |
| Dimiza et al. (2008) | Apr (2002), Aug (2001 and 2002) | Mediterranean sea | SEM | YES | 190 |
| Dimiza et al. (2015) | Jan (2007), Feb (2012) | Mediterranean sea | SEM | YES | 99 |
| | Mar (2002), Apr (2006) | | | | |
| | May (2013), Aug (2001) | | | | |
| | Sep (2004) | | | | |
| Eynaud et al. (1999) | Feb to Mar (1995) | South Atlantic Ocean | LM/SEM | NO | 40 |
| Giraudeau et al. (2016) | Aug to Sep (2014) | Barents Sea | pLM/SEM | YES | 170 |
| Godrijan et al. (2018) | Twice a month (2008-2009) | Mediterranean sea | LM/SEM | YES | 24 |
| Guerreiro et al. (2013) | Mar (2010) | Nazare Canyon, Portugal | pLM/SEM | YES | 108 |
| Guptha et al. (1995) | Sep to Oct (1992) | Arabian Sea | SEM | YES | 18 |
| Haidar and Thierstein (2001) | Jan 1991 to Jan 1994 | Bermuda, North Atlantic | pLM/SEM | YES | 217 |
| Karatsolis et al. (2017) | Oct (2013), Mar (2014) | Mediterranean sea | SEM | YES | 72 |
| | Oct (2013), Jul (2014) | | | | |
| Kinkel et al. (2000) | Aug to Sep (1994) | Atlantic ocean | SEM | NO | 47 |
| | Mar to Apr (1996) | | | | |
| | Jan to Mar (1997) | | | | |
| Luan et al. (2016) | Oct to Nov (2013) | Yellow and East China Seas | SEM | YES | 57 |
| Malinverno (2003) | Nov to Dec (1997) | Mediterranean sea | pLM/SEM | NO | 72 |
| Malinverno et al. (2015) | Jan (2001) | Southern Ocean, West Pacific | pLM/SEM | NO | 13 |
| Patil et al. (2017) | Jan to Feb (2010) | Southern Ocean | SEM | NO | 48 |
| Poulton et al. (2017) | May to Jun (2003) | Atlantic ocean | SEM | YES | 143 |
| | Apr to Jun (2004) | | | | |
| | Sep to Oct (2004) | | | | |
| | Oct to Nov (2005) | | | | |
| Saavedra-Pellitero et al. (2014) | Nov (2009) to Jan (2010) | Southern ocean | SEM | NO | 150 |
| Schiebel et al. (2011) | Mar (2004) | North Atlantic Ocean | SEM | NO | 47 |
| Schiebel et al. (2004) | May to Jun (1997) | Arabian Sea | SEM | YES | 49 |
| | and Jul to Aug (1995) | | | | |
| Smith et al. (2017) | Jan to Feb (2011), | Southern Ocean | SEM | NO | 27 |
| | Feb to Mar (2012) | | | | |
| Šupraha et al. (2016) | Feb (2013) and Jul (2013) | Mediterranean sea | SEM | YES | 63 |
| Takahashi and Okada (2000) | Feb to Mar (1996) | SE Indian Ocean | SEM | NO | 118 |
| Triantaphyllou et al. (2018) | Mar (2017) | Mediterranean sea | LM/SEM | YES | 42 |
| | Mar (2017) | | | | |
| Silver (2009) | Jan (2004) to Jun (2004) | Pacific Ocean (HOT) | SEM | NO | 13 |

**Table 2.** Taxonomic units included in HOLP-index

| Heterococcolithophores | Holococcolithophores |
|---|---|
| *C. mediterranea* | *C. mediterranea* HOL |
| *S. pulchra* | *S. pulchra* HOL |
| *S. protrudens* | |
| *S. bannockii* | *S. bannockii* HOL |
| *S. nana* | *S. nana* HOL |
| *S. arethusae* | *S. arethusae* HOL |
| *S. nodosa* | *H. cornifera* |
| *S. histrica* | *S. histrica* HOL |
| *S. molischii* | *S. molischii* HOL |
| *S. anthos* | *S. anthos* HOL |
| *S. strigilis* | *S. strigilis* HOL |
| *S. halldalii* | *S. halldalii* HOL |
| *S. marginiporata* | *S. marginiporata* HOL |
| *S. apsteinii* | *S. apsteinii* HOL |
| *P. japonica* | *P. japonica* HOL |
| *H. carteri* | *H. carteri* HOL |
| *H. wallichii* | *H. wallichii* HOL |
| *H. pavimentum* | *Helicosphaera* HOL *dalmaticus* type |
| *A. quattrospina* | *A. quattrospina* HOL |
| *A. robusta* | *S. quadridentata* |
| *R. clavigera* | |
| *R. xiphos* | |
| *C. aculeata* | *C. heimdaliae* |
| *C. leptoporus* | *C. leptoporus* HOL |
| *C. pelagicus* | *C. pelagicus* HOL |
| *C. quadriperforatus* | *C. quadriperforatus* HOL |
| *C. sphaeroidea* | *C. sphaeroidea* HOL |
| *P. arctica* | *P. arctica* HOL |
| *P. sagittifera* | *P. sagittifera* HOL |
| *P. borealis* | *P. borealis* HOL |
| *B. virgulosa* | *B. virgulosa* HOL |

Taxonomic units included in HOLP-index. Note that in some instances multiple heterococcolithophores are associated with single holococcolithophores (e.g. *S. pulchra* and *S. protrudens* are both associated with *S. pulchra* HOL

**Table 3.** Global hetero- and holococcolithophore abundance

| location | season | phase | mean (±ci) (cells L$^{-1}$) | max (cells L$^{-1}$) | contribution (±ci)(%) | n |
|---|---|---|---|---|---|---|
| Global | Mar-May | HET | 4.57e+04 (±4.72e+03) | 4.93e+05 | 93.72 (±0.98) | 585 |
| | | HOL | 2.00e+03 (±5.42e+02) | 8.72e+04 | 5.05 (±0.93) | 585 |
| | Jun-Aug | HET | 4.36e+04 (±1.56e+04) | 4.37e+06 | 82.53 (±1.68) | 739 |
| | | HOL | 4.64e+03 (±1.03e+03) | 2.23e+05 | 16.16 (±1.68) | 739 |
| | Sep-Nov | HET | 1.75e+04 (±3.09e+03) | 3.53e+05 | 91.46 (±1.61) | 438 |
| | | HOL | 1.74e+03 (±8.27e+02) | 1.27e+05 | 7.11 (±1.59) | 438 |
| | Dec-Feb | HET | 9.32e+04 (±8.99e+03) | 1.64e+06 | 95.37 (±0.66) | 772 |
| | | HOL | 1.78e+03 (±4.76e+02) | 1.18e+05 | 1.67 (±0.37) | 772 |
| Arctic Circle | Jun-Aug | HET | 5.83e+04 (±4.53e+04) | 4.37e+06 | 95.87 (±2.12) | 213 |
| | | HOL | 1.83e+03 (±2.18e+03) | 2.23e+05 | 3.71 (±2.02) | 213 |
| | Sep-Nov | HET | 3.41e+04 (±2.51e+04) | 1.29e+05 | 94.79 (±5.53) | 11 |
| | | HOL | 2.55e+02 (±2.71e+02) | 1.12e+03 | 5.21 (±5.53) | 11 |
| East China Sea | Sep-Nov | HET | 2.99e+04 (±1.30e+04) | 2.39e+05 | 96.48 (±3.97) | 51 |
| | | HOL | 9.06e+02 (±7.98e+02) | 1.47e+04 | 3.52 (±3.97) | 51 |
| Indian Ocean | Mar-May | HET | 1.13e+05 (±2.97e+04) | 2.18e+05 | 96.88 (±1.3) | 33 |
| | | HOL | 1.41e+03 (±7.85e+02) | 1.10e+04 | 2.35 (±1.31) | 33 |
| | Jun-Aug | HET | 2.40e+04 (±7.38e+03) | 1.11e+05 | 90.11 (±3.53) | 53 |
| | | HOL | 6.57e+02 (±2.67e+02) | 3.43e+03 | 3.68 (±2.09) | 53 |
| | Sep-Nov | HET | 7.03e+03 (±1.50e+03) | 3.33e+04 | 89.33 (±3.78) | 89 |
| | | HOL | 2.87e+02 (±2.00e+02) | 5.63e+03 | 5.57 (±3.49) | 89 |
| | Dec-Feb | HET | 2.00e+05 (±1.71e+03) | 2.27e+05 | 96.56 (±0.64) | 102 |
| | | HOL | 4.80e+03 (±1.27e+03) | 3.10e+04 | 2.3 (±0.6) | 102 |
| Mediterranean Sea | Mar-May | HET | 2.80e+04 (±4.90e+03) | 2.11e+05 | 88.88 (±3.19) | 146 |
| | | HOL | 3.76e+03 (±1.83e+03) | 8.72e+04 | 10.55 (±3.06) | 146 |
| | Jun-Aug | HET | 1.21e+04 (±1.95e+03) | 1.00e+05 | 68.42 (±2.92) | 290 |
| | | HOL | 9.42e+03 (±1.94e+04) | 1.02e+05 | 31.38 (±2.93) | 290 |
| | Sep-Nov | HET | 1.70e+04 (±4.38e+03) | 3.53e+05 | 89.11 (±2.89) | 195 |
| | | HOL | 3.10e+03 (±1.82e+03) | 1.27e+05 | 10.2 (±2.91) | 195 |
| | Dec-Feb | HET | 4.23e+04 (±1.18e+04) | 3.96e+05 | 96.78 (±1.11) | 125 |
| | | HOL | 2.21e+03 (±2.33e+03) | 1.18e+05 | 1.96 (±0.92) | 125 |
| Atlantic Ocean | Mar-May | HET | 4.20e+04 (±5.68e+03) | 1.83e+05 | 96.78 (±0.87) | 174 |
| | | HOL | 1.20e+03 (±5.36e+02) | 2.76e+04 | 1.88 (±0.6) | 174 |
| | Jun-Aug | HET | 1.51e+05 (±6.84e+04) | 1.55e+06 | 93.6 (±2.07) | 86 |
| | | HOL | 1.70e+03 (±8.96e+02) | 2.29e+04 | 3.89 (±1.6) | 86 |
| | Sep-Nov | HET | 1.48e+04 (±4.56e+03) | 5.40e+04 | 96.76 (±1.34) | 30 |
| | | HOL | 3.77e+02 (±1.54e+02) | 1.39e+03 | 2.59 (±1.22) | 30 |
| | Dec-Feb | HET | 2.50e+04 (±9.70e+03) | 9.78e+04 | 94.23 (±3.09) | 29 |
| | | HOL | 3.38e+02 (±1.55e+02) | 1.64e+03 | 1.81 (±1.26) | 29 |
| Pacific Ocean | Mar-May | HET | 1.43e+04 (±6.24e+03) | 4.96e+04 | 92.33 (±6.63) | 25 |
| | | HOL | 4.50e+03 (±4.56e+03) | 3.98e+04 | 7.55 (±6.64) | 25 |
| | Jun-Aug | HET | 1.96e+04 (±3.17e+04) | 1.48e+05 | 98.12 (±1.64) | 9 |
| | | HOL | 4.90e+01 (±9.70e+01) | 4.45e+02 | 0.03 (±0.07) | 9 |
| | Dec-Feb | HET | 2.00e+04 (±1.32e+04) | 1.64e+05 | 94.85 (±4.02) | 28 |
| | | HOL | 8.65e+02 (±1.20e+03) | 1.70e+04 | 0.89 (±0.75) | 28 |
| Southern Ocean | Mar-May | HET | 1.17e+05 (±2.88e+04) | 4.93e+05 | 98.19 (±0.88) | 50 |
| | | HOL | 9.10e+02 (±6.55e+02) | 1.60e+04 | 1.36 (±0.87) | 50 |
| | Dec-Feb | HET | 9.10e+04 (±1.66e+04) | 1.64e+06 | 99.05 (±0.7) | 332 |
| | | HOL | 3.24e+02 (±2.06e+02) | 2.67e+04 | 0.95 (±0.7) | 332 |

Values in parentheses are 95% confidence intervals.

**Table 4.** Global HOLP-index

| location | season | mean | n |
|---|---|---|---|
| Global | Mar-May | 17.33 ($\pm$2.55) | 332 |
| Global | Jun-Aug | 27.41 ($\pm$2.67) | 484 |
| Global | Sep-Nov | 18.29 ($\pm$3.84) | 241 |
| Global | Dec-Feb | 5.65 ($\pm$1.71) | 257 |
| Arctic Circle | Jun-Aug | 13.01 ($\pm$5.44) | 107 |
| East China Sea | Sep-Nov | 17.06 ($\pm$8.25) | 40 |
| Indian Ocean | Mar-May | 4.7 ($\pm$4.49) | 16 |
| Indian Ocean | Jun-Aug | 15.78 ($\pm$9.55) | 26 |
| Indian Ocean | Sep-Nov | 26.42 ($\pm$12.05) | 51 |
| Mediterranean Sea | Mar-May | 25.68 ($\pm$4.28) | 140 |
| Mediterranean Sea | Jun-Aug | 39.03 ($\pm$3.23) | 285 |
| Mediterranean Sea | Sep-Nov | 16.84 ($\pm$4.11) | 123 |
| Mediterranean Sea | Dec-Feb | 7.23 ($\pm$2.97) | 97 |
| Atlantic Ocean | Mar-May | 10.05 ($\pm$3.04) | 116 |
| Atlantic Ocean | Jun-Aug | 6.13 ($\pm$4.91) | 48 |
| Atlantic Ocean | Sep-Nov | 0.59 ($\pm$0.81) | 12 |
| Atlantic Ocean | Dec-Feb | 0.47 ($\pm$0.65) | 19 |
| Pacific Ocean | Mar-May | 38.2 ($\pm$20.58) | 15 |
| Pacific Ocean | Dec-Feb | 34.85 ($\pm$17.08) | 12 |
| Southern Ocean | Dec-Feb | 0.61 ($\pm$0.58) | 40 |

Mean HOLP-indices grouped by season and location. Values in parentheses are 95% confidence interval.

**Table 5.** PCA loadings of Mediterranean Sea data set

| Variable | PC1 | PC2 |
|---|---|---|
| Temperature | 1.57 | -0.75 |
| Salinity | -1.25 | 1.23 |
| Fixed Nitrogen | -0.42 | -1.04 |
| Silicate | -0.98 | -1.27 |
| Phosphate | -0.81 | -0.89 |
| DayLength | 1.17 | 0.27 |

The first two axis of the PCA captured
53.94% of variance. Data from Cerino et al.
(2017) and Godrijan et al. (2018)

**Table 6.** Spearman correlations for AMT data set

| Phase | Temp | Sal | PO4 | NOx | Depth | Si |
|---|---|---|---|---|---|---|
| HET | -0.095 | -0.085 | **-0.298*** | -0.095 | **-0.323\*\*\*** | -0.139 |
| HET.P | 0.13 | 0.136 | -0.069 | -0.092 | **-0.384\*\*\*** | **-0.295\*\*** |
| HOL | **0.339\*\*\*** | **0.224\*\*** | **-0.327*** | **-0.609\*\*\*** | **-0.584\*\*\*** | **-0.52\*\*\*** |
| HOL.P | **0.327\*\*\*** | **0.233\*\*** | **-0.289*** | **-0.55\*\*\*** | **-0.58\*\*\*** | **-0.502\*\*\*** |
| HOLP | **0.31\*\*\*** | **0.236\*\*** | **-0.506\*\*\*** | **-0.587\*\*\*** | **-0.472\*\*\*** | **-0.469\*\*\*** |

*** $p < 0.001$, ** $p < 0.01$, * $p < 0.05$

Significant correlations are highlighted in bold. Data was acquired from Poulton et al. (2017).

**Table 7.** Spearman correlations for Mediterranean data set

| Phase | Temp | Sal | PO4 | NOx | DayLen | Si |
|-------|------|-----|-----|-----|--------|-----|
| HET | **-0.304**** | **0.324**** | **0.213*** | **0.351***** | **-0.329**** | **0.373***** |
| P.HET | 0.096 | 0.18 | 0.08 | -0.009 | 0.029 | **0.208*** |
| HOL | **0.443***** | **-0.365***** | -0.071 | **-0.295**** | **0.475***** | -0.155 |
| P.HOL | **0.359***** | -0.056 | 0.042 | -0.079 | **0.357***** | 0.029 |
| HOLP | **0.418***** | -0.145 | 0.018 | -0.063 | **0.399***** | -0.031 |

*** $p < 0.001$, ** $p < 0.01$, * $p < 0.05$

Significant correlations are highlighted in bold. Data was acquired from Godrijan et al. (2018) and Cerino et al. (2017)

**Table 8.** Niche expansion and niche overlap.

| Species | Study | NE HET | NE HOL | Jaccard | Sørensen |
|---|---|---|---|---|---|
| Paired species | AMT | 0.11 | 0.05 | 0.84 | 0.91 |
| | BATS | 0.49 | 0.02 | 0.50 | 0.66 |
| | Med | 0.31 | 0.15 | 0.54 | 0.70 |
| *A. quattrospina* | AMT | 0.50 | 0.05 | 0.45 | 0.62 |
| | Med | 0.47 | 0.18 | 0.35 | 0.52 |
| *C. leptoporus* | AMT | 0.21 | 0.45 | 0.34 | 0.51 |
| | Med | 0.26 | 0.61 | 0.13 | 0.23 |
| *C. mediterranea* | AMT | 0.06 | 0.46 | 0.48 | 0.65 |
| | Med | 0.22 | 0.42 | 0.37 | 0.54 |
| *H. carteri* | AMT | 0.41 | 0.30 | 0.29 | 0.45 |
| *H. wallichii* | AMT | 0.42 | 0.47 | 0.11 | 0.20 |
| *S. anthos* | AMT | 0.69 | 0.04 | 0.27 | 0.42 |
| | BATS | 0.41 | 0.28 | 0.32 | 0.48 |
| *S. arethusa* | Med | 0.26 | 0.29 | 0.45 | 0.62 |
| *S. bannockii* | AMT | 0.17 | 0.19 | 0.63 | 0.77 |
| *S. halldalii* | AMT | 0.17 | 0.08 | 0.74 | 0.85 |
| *S. histrica* | AMT | 0.36 | 0.17 | 0.47 | 0.64 |
| | Med | 0.03 | 0.78 | 0.19 | 0.32 |
| *S. molischii* | AMT | 0.44 | 0.32 | 0.24 | 0.39 |
| | Med | 0.47 | 0.53 | 0.00 | 0.00 |
| *S. pulchra* | AMT | 0.18 | 0.14 | 0.68 | 0.81 |
| | BATS | 0.49 | 0.21 | 0.30 | 0.46 |
| | Med | 0.51 | 0.16 | 0.33 | 0.49 |
| *S. nana* | AMT | 0.50 | 0.06 | 0.44 | 0.62 |
| *S. strigilis* | Med | 0.12 | 0.53 | 0.35 | 0.52 |

Niche overlap and niche expansion metrics utilized in this study. For definitions see Sect 2.5 and Fig. 3. NE = Niche expansion