# Peer review of "The haplo-diplontic life cycle expands niche space of coccolithophores"

_Biogeosciences, 2020_

## Referee Comment (RC1) · Anonymous Referee #1 · 29 Jul 2020

This is a beautifully crafted review drawing attention to the overlooked global importance of coccolithophore species for which heterococcolithophore (diploid)-holococcolithophore (haploid) pairing has been identified. As stated, roughly these paired species account for 18% of total coccolithophore abundance, but because the diploid phase tends to be more heavily calcified than the lighter, or non-calcified, haploid phase, they are likely to occupy different ecological niches, as exemplified by different biogeography (Fig.3; eg. holococcolithophores rare in Southern Ocean >50oS), different depth profiles (Fig.4; holococcolithophores in upper 50 m, heterococcolithophores evenly distributed with depth), different seasonality (Fig.7; but apparently different niches off Bermuda and in Mediterranean; this needs to be better explained). The coloured SEM plate in Fig.2 is spectacular and I applaud how the red (diploid) and

blue (haploid) colour coding is consistently adopted throughout all graphs.

What knowledge is missing, albeit repeatedly admitted, is how this haplo-diplontic life cycle works to expand the niche for the globally most abundant (59.2%) coccolithophore Emiliania huxleyi, because its non-calcified haploid stage cannot currently be identified by regular LM and which calls for new molecular techniques [1].

A number of these knowledge gaps where future work should focus, best should be summarised in the abstract. This also includes [2] more SEM observations in the Pacific; [3] role of carbonate chemistry within the haploid-diploid niche space; and [4] resolution of the fact that grouped hetero-holococcolithophore abundances may not always be the best representation for individual species.

In detail: How was the estimate made that the haplo-diplontic life cycle expands the coccolithophore niche space by 17%??

―――――――――――――――

---

## Referee Comment (RC2) · Anonymous Referee #2 · 2 Aug 2020

The study by de Vries et al. is a broad synthesis studies, which a main focus on describing environmental portioning and drivers behind differential haplo-diplontic stages of the coccolithophores. As the haploid stage is often overlooked, yet it is ecologically and biogeochemically important, this is an important review studies of the inclusion of the life stages towards an improved understanding of the coccolithophore ecology.

While the paper joins various components of coccolithophores biology/ecology, my major questions are focus on the methodological approach, which in many ways is insufficiently described/explained, with some of the resulting conclusions then also not supported. I would like the authors to address the following methodological approaches.

There are several caveats behind such synthesis approach that need to be highlighted and further elaborated. The first and most important is compilation of SEM dataset,

which strikes me as uncertain (what is the images were not taken) and difficult to present in the quantitative terms. Were both phases in the initial SEM dataset presented quantitatively or where there also just studies that took qualitative SEM approach? How did this impact how the authors proceed with the study? Where are potential biases? How can you estimate uncertainty in the quantified approach?

Second, I do follow the nice overlap and nice expansions in the hyperspace. It is not clear to me how the authors transitioned from hypervolume to the nice space and how are the two haplo-diplontic stages represented in the Eq 2 (line 15) based on the similarity metrics? How was the intersection or the union between two hypervolumes determined? Why was NE not calculated for BATS?

Third, in the section of seasonality (line 135), only a handful of environmental parameters are missing and there could be other important physical-chemical drivers. For example, In Figure 8, you also include turbulence in there, why was such parameter not included in the PCA analyses? What about pH, for example? Also, as described are large-scale patterns, what about mesoscale type events, advection and other physical parameters? In line 210, why were not the same approaches used for the Med and ATM? By using water column vs niche space approach, this excludes the possibility of comparing two regions.

Forth, where is 17% of expanding niche space coming from?

Fifth, the authors report 7.3 to 18% of the species abundance, which is a relatively wide range and needs to be better quantified with the uncertainty. Also, given the quantitative estimates presented, I wish the authors to have better addressed some of the knowledge gaps, the impact on the pump of the haploid-diploid stage, standardizing the approaches to represent different species (paired- non paired, etc),. Based in Figure 3, one could conclude that the relative abundances (f,g,h) of holococcos are only slightly lower compared to the heterococcos (c,d,e) and these figures need changes. What is the difference in shading?

---

## Referee Comment (RC3) · Anonymous Referee #3 · 18 Aug 2020

General comments This is an interesting paper, timely, and relevant to the field of physiological ecology of phytoplankton. It deserves to be published but needs some minor revision. The paper was a bit sloppy in spots, with a number of typos. The paper should be checked over carefully prior to final submission.

There are some terms that really need to be clarified in the revision to avoid confusion and to sharpen their points. First, when discussing nitrogenous nutrients they refer to "fixed nitrogen" as the sum of nitrate and nitrite (line 90-91). This reviewer has no idea why they are using the adjective "fixed" for the sum of these molecules (and they do not include ammonium or urea in that sum, for example). Typically, the fixation of nitrogen by phytoplankton is describing the uptake and assimilation of N2 gas into organic nitrogen fractions, which is not what they are describing. I would advocate that

they globally scrub the term "fixed nitrogen" and replace it with something like dissolved inorganic nitrogen (DIN, here defined as nitrate + nitrite only). Second, in their equation about niche expansion (line 150) they refer to terms describing the "intersection of hypervolumes" and the "union of hypervolumes". If there is a union of hypervolumes, then they also intersect, right? The authors must very carefully define the difference between these. As long as there are ambiguities in the definition of those terms, then the entire niche expansion argument won't have much relevance.

Finally, they talk about a 7% contribution of holococcolithophore abundance to the total coccolithophore abundance as being significant (abstract and line 331). It may be statistically significant, but it seems to this reviewer to be a little overblown. I would suggest that holococcolithophores more appropriately would be considered a minor constituent of the total coccolithophore assemblage. For holo/heterococcolith paired species, the holococcolithophore abundance represents ∼18% of the paired species abundance, only about a fifth, definitely still a minor fraction, at best. This doesn't detract from the results. It is still a fascinating observation and the question that arises to this reviewer is why is that fraction so small? This paper requires some revision but it provides new insights to a very real problem in coccolithophore ecology. It deserves to be published and will be cited well. The authors simply need to clean it up a bit.

Specific comments Line 4 after "diploid life cycle phase" are they referring to coccolithophores only or other organisms. Please clarify.

Line 13 "ballast" not "ballasts"

Line 13-15 They are describing the biological carbon pump, not the carbonate pump (aka alkalinity pump). The linkage of calcite production to the biological carbon pump is a strong one via ballasting of organic carbon to the sea floor. This is not the alkalinity pump however. Klass and Archer (2002) were looking at the impact of ballasting of sinking POC and the effect on the rain ratio.

Line 16 Globally, about one quarter of all marine sediments are calcium carbonate.

Citing the 30-90% value presents a skewed view of the importance of calcite sediments on Earth.

Line 18- Given that this sentence is going back to the biological carbon pump, you might move it up in the paragraph where you are first mentioning the biological carbon pump.

Line 30 add an "s" . . ."A few organism"

Line 91 Reference to "fixed nitrogen" and all subsequent uses of that term in the paper. . .see general comments above.

Lines 151 and 152- Must describe how the "union" and "intersection" of hypervolumes are being distinguished. See general comments above.

Line 162- Again, they are describing the function in R to calculate the "intersection" of hypervolumes when the reader may not be clear about the difference between this calculation and that of the union of the hypervolumes! This is a really important distinction.

Line 185 As they state, on a regional basis, holococcolithophores generally contributed <6% of total coccolithophore abundance. This seems pretty minor to be honest!

Line 190- change to . . ."where a HOLP-index". . . not "an Holp-index"

Line 193- add comma, "In the global data set, heterococcolithophore. . ."

Line 218- change to . . ."high nutrient concentrations, cold water temperatures at depth or other factors not addressed in this study".

Line 253 They show significant positive correlations with silicate. This is a very interesting observation. The Discussion section should have a few sentences explaining how this could be!

Line 273- add "s" to heterococcolithophore to make it plural.

Line 285 add comma, "overlap metrics, respectively"

Line 331 "Our meta-analysis shows that holococcolithophores are important contributors to coccolithophore abundance and ecology contributing ∼7.3% to total coccolithophore abundance" This observation doesn't match the data. 7.3% is a small number. Call it like it is!

Line 336- re-word this so that it agrees with the minor contribution…"past and future oceans could have other biogeochemical effects. A shift towards"…

Line 363- Remove "?"

Line 366- improper hyphenation of wrap-around word "coccolithophore"

Line 370- I disagree with this statement. Calcification measurements are including the calcite production of holo- and heterococcoliths. However, the standing stock of calcite is being underestimated by not including the holococcolithophore abundance. Also, leave off the last words of the sentence, "or activity".

Line 380- Sentence "Overall observations in the haploid stage of E. huxleyi are…". There is some classic literature that the authors should cite from the mid 1990's: Campbell, L., et al. (1994). "Immunochemical characterization for eukaryotic ultraplankton from the Atlantic and Pacific oceans." Journal of Plankton Research 16(1): 35-51. They used immunochemical antisera to identify haploid stages of E. huxleyi. Line 385; Again, there were a number of classic SEM studies from the Pacific Ocean. One by Reid (1980). Reid, F. (1980). "Coccolithophorids of the North Pacific Central Gyre with notes on their vertical and seasonal distribution." Micropaleontology 26: 151-176. The SEM plates in the paper are meticulous and it might be worth a look before you discount all Pacific SEM observations. See also previous work of Honjo and Okada from the Pacific.

Line 404- There is contrary evidence you should cite to be balance, though: Rivero-Calle, S., et al. (2015). "Multidecadal increase in North Atlantic coccolithophores and

the potential role of rising CO2." Science 350(6267): 1533-1537.

Line 407- eliminate the "s" from "compositions"

Line 414- Reword, "Our analysis shows that holococcolithophores constitute about one fifth of total paired coccolithophore abundance. . ."

Figures:

Figures 3 and 4- The font on all the axes is way to small to be readable. These must be increased in size.

Figure 4- Legend is reversed for red and blue colors. . .Heterococcos are plotted in red (not blue) and holococcos are plotted in blue (not red).

Figures 5 and 6- No units are provide in this figure or the legend for the color bars!

Fig. 6 change "fixed nitrogen" to DIN (see also Fig. 7)

Table 2 is excellent and a great reference. Should you state the names for the holo forms of R. clavigera and R. xiphos since you have left them blank?

Tables 6 and 7- You never discuss the significant relationships with Silicate (not "Si" as you say in the table!) This really deserves some discussion.

Tables 8 and 9 The legends are very minimalistic. Please move your definition of NE1 and NE2 to the legend from the footnotes. This needs to be more obvious to the reader. Also, maybe specify in the table legend what the Jaccard and Sorensen columns refer to (and units?) or refer the reader to the text.

---

## Author Comment (AC1) · 27 Aug 2020

**BG Discussion: Reply to all reviewers**

We would like to thank all reviewers for their constructive feedback. Based on their suggestions we will make the following changes:

- Update the niche expansion noted in the abstract and conclusion from 17% to 18.8% (the mean for paired species) as well as the range of values observed for individual species (3-76%).
- Update the axis labels of latitudinal plots in Fig. 3.
- Update Fig. 7 to include both time and depth.
- Include PCA plots of the BATS, Med, and AMT data sets within temperature, salinity, and nitrogen as supplementary figures.
- Include niche expansion analysis for the BATS station.
- Clarify that holococcolithophores generally constitute a minor component of the total coccolithophore abundance but that holococcolithophores dominate under certain environments and are furthermore important in terms of niche space.
- 15
- Clarify the hypervolume metrics with visual representation.
- Discuss why contribution of holococcolithophores is minor.
- Discuss importance of time and depth in structuring niche space.
- Discuss positive Spearman correlations with silicate.
- Discuss uncertainties and biases of the SEM compilation.
  - Discuss impact of physical processes.
  - Discuss gaps of knowledge in dedicated section.

We provide a response to the reviewers and a detailed explanation of our changes below.

**25 BG Discussion: Reply to RC1**

We would like to thank the first anonymous reviewer for the kind and positive feedback. A detailed response to their comments is found below.

This is a beautifully crafted review drawing attention to the overlooked global importance of coccolithophore species for which heterococcolithophore 30 (diploid)-holococcolithophore (haploid) pairing has been identified. As stated, roughly these paired species account for 18% of total coccolithophore abundance, but because the diploid phase tends to be more heavily calcified than the lighter, or non-calcified, haploid phase, they are likely to occupy different

10

 $\mathbf{5}$

ecological niches, as exemplified by different biogeography (Fig.3; eg. holococ-

- 35 colithophores rare in Southern Ocean>50S), different depth profiles (Fig.4; holococcolithophores in upper 50 m, heterococcolithophores evenly distributed with depth), different seasonality (Fig.7; but apparently different niches off Bermuda and in Mediterranean; this needs to be better explained).
- The apparent different niches off Bermuda and in the Mediterranean, as 40 observed in Fig. 7, were the result of averaged depth profiles, which squeezed together the temporal and spatial pattern. We have now updated the original figure (which was plotted to time) to include both time and depth (see Figure 1 attached). This figure shows depth to be an important variable and that holococcolithophores are predominantly present in low-nutrient regions, which
- 45 is consistent with our Spearman correlations (Table 6 and Table 7 of the original manuscript) and observed PCA niche space (Fig. 6 of the original manuscript). We will replace Fig. 7 with this new figure in the manuscript.

The coloured SEM plate in Fig. 2 is spectacular and I applaud how the red (diploid) and blue (haploid) colour coding is consistently adopted throughout 50 all graphs.

What knowledge is missing, albeit repeatedly admitted, is how this haplodiplontic life cycle works to expand the niche for the globally most abundant (59.2%) coccolithophore *Emiliania huxleyi*, because its non-calcified haploid stage cannot currently be identified by regular LM and which calls for new molecular techniques [1].

A number of these knowledge gaps where future work should focus, best should be summarised in the abstract. This also includes [2] more SEM observations in the Pacific; [3] role of carbonate chemistry within the haploid-diploid niche space; and [4] resolution of the fact that grouped hetero-holococcolithophore abundances may not always be the best representation for individual species.

We will update our discussion to include a section to specifically highlight the current knowledge gaps mentioned above, as well as some gaps raised by the second reviewer below [line 195]. This discussion will cover mainly aspects on the impact on the carbon pump and the limited descriptions of the life cycle pairs.

In detail: How was the estimate made that the haplo-diplontic life cycle expands the coccolithophore niche space by 17%??

Thank you for pointing out the lack of description for the 17% estimate of niche space expansion. This number was from an older version of the manuscript

70 (which used a different statistical analysis). We will correct this number with the current analysis including the range observed in the Mediterranean, BATS, and AMT data sets (e.g. 3-78%), as well as the average NE observed for paired species (18.8%).

**BG Discussion: Reply to RC2**

55

60

65

75 We would like to thank the second anonymous reviewer for the in-depth and constructive feedback. A detailed response to questions raised are provided

**below.**

95

115

120

The study by de Vries et al. is a broad synthesis studies, which a main focus on describing environmental portioning and drivers behind differential haplo-diplontic stages of the coccolithophores. As the haploid stage is often overlooked, yet it is ecologically and biogeochemically important, this is an important review studies of the inclusion of the life stages towards an improved understanding of the coccolithophore ecology.

While the paper joins various components of coccolithophores biology/ecology,
my major questions are focus on the methodological approach, which in many ways is insufficiently described/explained, with some of the resulting conclusions then also not supported. I would like the authors to address the following methodological approaches.

There are several caveats behind such synthesis approach that need to be 90 highlighted and further elaborated.

The first and most important is compilation of SEM dataset, which strikes me as uncertain (what is the images were not taken) and difficult to present in the quantitative terms. Were both phases in the initial SEM dataset presented quantitatively or where there also just studies that took qualitative SEM approach? How did this impact how the authors proceed with the study?

Where are potential biases? How can you estimate uncertainty in the quantified approach?

We agree that the SEM compilation represents a certain degree of uncertainty, and as such the resulting analysis primarily serves as a first order esti-

100 mate of global coccolithophore abundance. The degree of uncertainty is noted in the large standard deviations observed in both the abundance and HOLP-index estimates. We will further clarify and discuss this in the manuscript.

We have limited uncertainty by focusing our analysis only on only one technique (SEM), which reduces uncertainties due to method comparison. Further-

105 more, SEM is more accurate in distinguishing life stages of coccolithophores than other microscopy techniques. (Bollmann et al., 2002; Cerino et al., 2017; Godrijan et al., 2018).

In addition, to account for identification uncertainty, we limited our HOLPindex analysis to studies that identified holococcolithophores to a species level,

110 and limited our in depth analysis (e.g. Mediterranean and Atlantic) to studies which we were confident accurately identified the samples.

There is likely a bias towards the Mediterranean Sea and the Atlantic Ocean due to the large number of samples in these regions. There is potentially also a temporal bias to bloom seasons. We will further develop these aspects on the level of uncertainties of our data set in the Methods and Discussion sections.

Second, I do follow the nice overlap and nice expansions in the hyperspace. It is not clear to me how the authors transitioned from hypervolume to the nice space and how are the two haplo-diplontic stages represented in the Eq 2 (line 15) based on the similarity metrics? How was the intersection or the union between two hypervolumes determined?

Hypervolume and niche space are interchangeable terms. We will update the manuscript to reflect this. We utilized the R package described in Blonder et al. (2014) and already referred in our manuscript to the original publication for the mathematical description of the hypervolume algorithms. The package calculates the 'niche space' hypervolume using kernel density estimations, and provides shape, volume (|A|and |B|), intersection, union and set difference of hypervolumes as well as similarity metrics. Since the method also resulted in some confusion for RC3 we will include Figure 3 (attached at end) in the hypervolume methods section. Why was NE not calculated for BATS?

125

130

135

145

We focused our analysis of environmental drivers on the AMT and Med data sets, and for consistency decided to do the same for NE. Furthermore, we have limited data to perform a NE at BATS, because BATS presents only two paired species for which both life cycle phases to be present. We will however include NE analysis for BATS in our updated version of the manuscript which

is provided as an attachment (Table 1). We find that the NE analysis of BATS has more similar values to the Mediterranean than the AMT data set, although one would expect them to be more similar to the latter based on hydro-graphic similarity. This indicates the potential importance of the nature of the data set, here a transect vs a time series,

140 tential importance of the nature of the data set, here a transect vs a time series which focus on spatial versus temporal correlations.

Third, in the section of seasonality (line 135), only a handful of environmental parameters are missing and there could be other important physical-chemical drivers. For example, In Figure 8, you also include turbulence in there, why was such parameter not included in the PCA analyses? What about pH, for example?

Turbulence was not measured in any of the original publications of our data set, and pH was only measured in some of the data sets. We have thus not included these variables in our analysis. We included turbulence in Figure 8

- 150 because it is commonly included in the Margalef niche space model (Margalef, 1978; Houdan et al., 2006; Frada et al., 2018). Although, as already argued in the manuscript, the relationship to turbulence broadly holds in terms of general patterns (MLD and seasonality), we agree that our data set does not explicitly support inclusion of turbulence of the model and we will update the 155 discussion to better reflect this. pH (and more specifically carbonate chemistry)
- is potentially a key factor on coccolithophore ecology as we discuss in lines 396-407. This discussion will be expanded to highlight the contradictory nature of pH effects as noted by reviewer 3 [line 366].
- Also, as described are large-scale patterns, what about mesoscale type events, advection and other physical parameters?

This is a good point. The effects of physical processes are difficult to constrain but this certainly warrants discussion. Effects of mesoscale processes should be partly negated since we averaged our data over several years. However, other physical processes may be important to consider. Godrijan et al.

165 (2018), for instance, noted that the East Adriatic Current (EAC) may be partly responsible for holococcolithophore abundance during winter and spring. We will add this to our discussion of the manuscript.

In line 210, why were not the same approaches used for the Med and ATM?

By using water column vs niche space approach, this excludes the possibility of comparing two regions.

We used different approaches for the two different data sets due to the different nature of the two data sets (AMT is spatial and Med is temporal). Although the two data sets can be compared by limiting the principal components of the PCA to salinity, temperature and salinity, the structure of the PCA plot is

175 strongly influenced by both depth and time (time is represented by day length in the Med). Not including these variable results in lack of separation between the two life cycle phases (see Fig. 4, Fig. 5, and Fig. 6 attached).

This in itself is interesting and we will discuss this in the Discussion as the importance of depth is also apparent in Fig. 7 (see discussion above [line 40]).

180 We will additionally include (Fig. 4, Fig. 5, and Fig. 6 attached as supplemental figures).

Forth, where is 17% of expanding niche space coming from?

Thank you for pointing out the lack of description for the 17% estimate of niche space expansion. Similarly to the comments from reviewer 1, this number was from an older version of the manuscript (which used a different statistical analysis). We will correct this number with the current analysis including the range observed in the Mediterranean, BATS, and AMT data sets (e.g. 3-78%), as well as the average NE observed for paired species (18.8%).

Fifth, the authors report 7.3 to 18% of the species abundance, which is a relatively wide range and needs to be better quantified with the uncertainty.

These are two different numbers. Calcifying haploid coccolithophores account for 7.3% of the total abundance, and 18% of the paired-species abundance. We understand the confusion and will clarify this in the manuscript.

Also, given the quantitative estimates presented, I wish the authors to have better addressed some of the knowledge gaps, the impact on the pump of the haploid-diploid stage, standardizing the approaches to represent different species (paired- non paired, etc),.

We are happy to discuss current knowledge gaps in further detail in the manuscript. The impact of the life cycle phases on the biological pump is cur-

200 rently not mentioned but critical. We will include a discussion in the knowledge gaps section. Standardization of paired and non-paired species is currently not possible as new HET-HOL pairs are still being described (discussed in page 4, line 110). We have followed the most up-to-date understanding of coccol-ithophore life cycle pairs from Frada et al. (2018), however this will change 205 in the future as more pairs are described. We will include this point in the

205 in the future as more p knowledge gaps section.

190

Based in Figure 3, one could conclude that the relative abundances (f,g,h) of holococcos are only slightly lower compared to the heterococcos (c,d,e) and these figures need changes.

210 This is a good point. We have updated the axis to absolute abundance and will provide the updated Fig. 3 below in the revised manuscript (see Fig. 2 attached).

What is the difference in shading?

The light shading on the latitudinal plots is log transformed (as noted in the caption). We have updated the axis to better reflect this (see Fig. 2 attached).

**BG Discussion: Reply to RC3**

We would like to thank the third reviewer for their positive and constructive feedback. Our in-depth response can be found below.

- General comments This is an interesting paper, timely, and relevant to the field of physiological ecology of phytoplankton. It deserves to be published but needs some minor revision. The paper was a bit sloppy in spots, with a number of typos. The paper should be checked over carefully prior to final submission. There are some terms that really need to be clarified in the revision to avoid confusion and to sharpen their points. First, when discussing nitrogenous
- 225 nutrients they refer to "fixed nitrogen" as the sum of nitrate and nitrite (line 90-91). This reviewer has no idea why they are using the adjective "fixed" for the sum of these molecules (and they do not include ammonium or urea in that sum, for example). Typically, the fixation of nitrogen by phytoplankton is describing the uptake and assimilation of N2 gas into organic nitrogen fractions,
- 230 which is not what they are describing. I would advocate that they globally scrub the term "fixed nitrogen" and replace it with something like dissolved inorganic nitrogen (DIN, here defined as nitrate + nitrite only).

We will replace 'Fixed nitrogen' with dissolved inorganic nitrogen (DIN) as suggested.

235 Second, in their equation about niche expansion (line 150) they refer to terms describing the "intersection of hypervolumes" and the "union of hypervolumes". If there is a union of hypervolumes, then they also intersect, right? The authors must very carefully define the difference between these. As long as there are ambiguities in the definition of those terms, then the entire niche expansion argument won't have much relevance.

In set theory the union includes all data points. While the difference is only the overlapping set of data points. We will include Figure 3 (attached) to the manuscript to clarify.

Finally, they talk about a 7% contribution of holococcolithophore abundance to the total coccolithophore abundance as being significant (abstract and line 331). It may be statistically significant, but it seems to this reviewer to be a little overblown. I would suggest that holococcolithophores more appropriately would be considered a minor constituent of the total coccolithophore assemblage. For holo/heterococcolith paired species, the holococcolithophore

250 abundance represents 18% of the paired species abundance, only about a fifth, definitely still a minor fraction, at best. This doesn't detract from the results. It is still a fascinating observation and the question that arises to this reviewer is why is that fraction so small?

We agree that 7% is a minor fraction of total coccolithophore abundance. It
is however not an insignificant fraction. Our argument is mainly that holococcolithophores are not insignificant rather than contributing significantly to total

 $\mathbf{6}$

coccolithophore abundance. We furthermore show that under certain circumstances holococcolithophores do become the dominant fraction. We will update our manuscript to reflect this.

As part of our discussion, we will comment on the reasons why the haploid fraction is much smaller than for the diploid life cycle phases:

- Strong dominance of *E. huxleyi*, which has a haploid naked phase
- Abundance of haploids in low nutrient regions which sustain a smaller total biomass compared to high nutrient regions (provided there are no other limiting factors).
- Potential sampling bias, specifically to bloom seasons when haploid abundance is low.

We will also highlight the implications towards calcium carbonate production (haploid tends to have a lower PIC:POC ratio) and how this will change under
future climate (potentially more haploid relative to diploid in stratified regions with reduced nutrient supply). A scenario which is projected to occur in each major ocean basin under Representative Concentration Pathway (RCP) 8.5 (Fu et al., 2016).

This paper requires some revision but it provides new insights to a very real 275 problem in coccolithophore ecology. It deserves to be published and will be cited well. The authors simply need to clean it up a bit.

Specific comments

Line 4 after "diploid life cycle phase" are they referring to coccolithophores only or other organisms. Please clarify.

Line 13 "ballast" not "ballasts"

Line 13-15 They are describing the biological carbon pump, not the carbonate pump(aka alkalinity pump). The linkage of calcite production to the biological carbon pump is a strong one via ballasting of organic carbon to the sea floor. This is not the alkalinity pump however. Klass and Archer (2002) were looking at the impact of ballasting of sinking POC and the effect on the

285

rain ratio.

280

That is a good point. We will further clarify the distinction between the carbonate and organic carbon pump and include a reference for the former (i.e. Zeebe (2012))

290 Line 16 Globally, about one quarter of all marine sediments are calcium carbonate. Citing the 30-90% value presents a skewed view of the importance of calcite sediments on Earth.

Line 18- Given that this sentence is going back to the biological carbon pump, you might move it up in the paragraph where you are first mentioning the biological carbon pump.

Line 30 add an "s"..."A few organism" Line 91 Reference to "fixed nitrogen" and all subsequent uses of that term in the paper...see general comments above.

Lines 151 and 152- Must describe how the "union" and "intersection" of hypervolumes are being distinguished. See general comments above.

265

260

300 Line 162- Again, they are describing the function in R to calculate the "intersection" of hypervolumes when the reader may not be clear about the difference between this calculation and that of the union of the hypervolumes! This is a really important distinction.

Line 185 As they state, on a regional basis, holococcolithophores generally contributed

Figure 1: Seasonality of hetero- and holococcolithophores at the BATS station in Bermuda (left column) and the RV-001 and LTER-1 stations in the Mediterranean Sea (right column). Note heterococcolithophores are most abundant in the winter followed by a high abundance of holococcolithophores in the late spring and early summer. (**a-b**) Heterococcolithophore abundance; (**b-c**) Holococcolithophore abundance; (**e, g**) Temperature; (**f, h**) Chlorophyll; (**i, k**) DIN (nitrite + nitrate); (**j, l**) Silicate.

---

## Author Response (AR1)

**BG Discussion: Reply to all reviewers**

We would like to thank all reviewers for their constructive feedback. We provide a response to the reviewers and a detailed explanation of our changes below.

**BG Discussion: Reply to RC1**

We would like to thank the first anonymous reviewer for the kind and positive feedback. A detailed response to their comments is found below.

This is a beautifully crafted review drawing attention to the overlooked global importance of coccolithophore species for which heterococcolithophore (diploid)-holococcolithophore (haploid) pairing has been identified. As stated, roughly these paired species account for 18% of total coccolithophore abundance, but because the diploid phase tends to be more heavily calcified than the lighter, or non-calcified, haploid phase, they are likely to occupy different ecological niches, as exemplified by different biogeography (Fig.3; eg. holococcolithophores rare in Southern Ocean>50S), different depth profiles (Fig.4; holococcolithophores in upper 50 m, heterococcolithophores evenly distributed with depth), different seasonality (Fig.7; but apparently different niches off Bermuda and in Mediterranean; this needs to be better explained).

The apparent different niches off Bermuda and in the Mediterranean, as was observed in Fig. 7 (now Fig. 8), was an artefact due to the averaged depth profiles, which squeezed together the temporal and spatial patterns. We have now updated the original figure (which was only plotted with time) to include both time and depth (see Fig. 8). We have also added the following changes to the text [line 389, page 13]:

> *Clear seasonal patterns are observed for DIN and silicate concentrations at both locations. High holococcolithophore abundances are observed when silicate and DIN concentrations are low, and vice versa for high heterococcolithophore abundances. This observed seasonal patterns at the BATS and Mediterranean time series are thus consistent with our Spearman correlations (see Table 6 and Table 7 and our discussion above) and PCA niche space (see Fig. 7 and discussion above).*

The coloured SEM plate in Fig. 2 is spectacular and I applaud how the red (diploid) and blue (haploid) colour coding is consistently adopted throughout all graphs.

What knowledge is missing, albeit repeatedly admitted, is how this haplo-diplontic life cycle works to expand the niche for the globally most abundant (59.2%) coccolithophore *Emiliania huxleyi*, because its non-calcified haploid stage cannot currently be identified by regular LM and which calls for new molecular techniques [1].

A number of these knowledge gaps where future work should focus, best should be summarised in the abstract. This also includes [2] more SEM observations in the Pacific; [3] role of carbonate chemistry within the haploid-diploid niche space; and [4] resolution of the fact that grouped hetero-holococcolithophore abundances may not always be the best representation for individual species.

We thank the reviewer for identifying these points. We have updated our abstract to the main knowledge gaps [line 15, page 1]:

> *Our compilation highlights the spatial and temporal sparsity of SEM measurements, and the need for new molecular techniques to identify uncalcified haploid coccolithophores. Our work also emphasizes the need for further work on the carbonate chemistry niche space of the coccolithophore life cycle.*

In detail: How was the estimate made that the haplo-diplontic life cycle expands the coccolithophore niche space by 17%??

We have now included a description of how the niche expansion was calculated [line 205, page 7]:

> *Niche overlap and niche expansion were calculated only for species for which both life cycle phases were observed. In addition to calculating the niche expansion for individual species, we calculated a average niche expansion by taking the mean NE values of the grouped species for each location for both the haploid and diploid coccolithophores life cycle phases.*

**BG Discussion: Reply to RC2**

We would like to thank the second anonymous reviewer for the in-depth and constructive feedback. A detailed response to questions raised are provided below.

The study by de Vries et al. is a broad synthesis studies, which a main focus on describing environmental portioning and drivers behind differential haplo-diplontic stages of the coccolithophores. As the haploid stage is often overlooked, yet it is ecologically and biogeochemically important, this is an important review studies of the inclusion of the life stages towards an improved understanding of the coccolithophore ecology.

While the paper joins various components of coccolithophores biology/ecology, my major questions are focus on the methodological approach, which in many ways is insufficiently described/explained, with some of the resulting conclusions then also not supported. I would like the authors to address the following methodological approaches.

There are several caveats behind such synthesis approach that need to be highlighted and further elaborated.

The first and most important is compilation of SEM dataset, which strikes me as uncertain (what is the images were not taken) and difficult to present in the quantitative terms. Were both phases in the initial SEM dataset presented quantitatively or where there also just studies that took qualitative SEM approach? How did this impact how the authors proceed with the study?

Where are potential biases? How can you estimate uncertainty in the quantified approach?

We appreciate the lack of information we originally presented on our SEM dataset. We have now added a section in the Methods to discuss the sources of bias and the implication of an incomplete data cover for our study [lines 120, page 5]:

**Sampling bias and cover of data set**

*Our compilation contains sampling bias and is spatially and temporally incomplete. Temporally, there is bias towards the months Jun-Aug and Dec-Feb (29.28% and 30.59% of samples, respectively), with fewer samples in the inter-seasons. This temporal bias results from generally higher sampling effort in the Arctic Circle in Jun-Aug (8.43% of samples) and the Southern Ocean in Dec-Feb (13.15% of samples). Not coincidentally, this is when and where coccolithophore abundances are the highest (see results below). When excluding the Arctic Circle (Jun-Aug) and the Southern Ocean (Dec-Feb), the data set is temporally relatively evenly distributed (28.20% Mar-May, 26.58% Jun-Aug, 22.13% Sep-Nov, 22.23% Dec-Feb).*

*Spatially, there is higher sampling in the Atlantic Ocean, Mediterranean Sea, Arctic Circle and Southern Ocean. In terms of spatial cover, coverage is limited in the Pacific Ocean and data is lacking in the Southern Ocean between Jun-Aug and the Arctic Circle between Dec-May. However, previous studies note the low coccolithophore abundance in the tropical and subtropical Pacific Ocean (Okada and Honjo, 1973; Honjo and Okada, 1974; Reid, 1980), and the absence or low abundance of holococcolithophores in this region (Okada and Honjo, 1973; Honjo and Okada, 1974; Reid, 1980). The lack of data in the Southern Ocean and the Arctic Circle for specific months is due to the difficulty of sampling these regions in the winter as well as low coccolithophore abundance due to light limitation.*

*The incomplete data cover of our data set combined with the spatial and temporal bias means that the analysis presented here mainly serves as a first order estimate of the relative hetero- and holococcolithophore abundance and distribution patterns. For more accurate estimates, additional sampling needs to be conducted.*

To better account for the uncertainty in our data set, we have replaced the standard deviations with confidence intervals. We have furthermore binned the results per season. These results have been added to Table 1, visualized in a new figure (Fig S1), and has been updated in the Methods section [line 112, page 4]:

*To reduce the effects of seasonality, we binned the data into four main seasons, defined as Dec-Feb, Mar-May, Jun-Aug, Sep-Nov. We also calculated on a global scale and regional scale the mean of the observed abundances and estimated the highest observed abundances (the 'maximum abundance') for both hetero- and holococcolithophores and each season. For the mean abundance calculations the mean was calculated for each sample and then averaged. Finally, we tested the count data for a normal distribution using a Shapiro-Wilks test for each region and the global data set. Where the count distribution was found normal (all data), a 95% confidence interval was calculated.*

We have also updated the results section with numbers updated based on season [line 217, page 8]:

*Highest maximum abundances of heterococcolithophores are observed at high latitudes within the Arctic circle ($>66°$ N) ($\approx 4.37$ x $10^6$ cells $l^{-1}$ for Jun-Aug), and the Southern Ocean ($>40°$ S and $<65°$ S) ($\approx 1.64$ x $10^6$ cells $l^{-1}$ for Dec-Feb). Generally, maximum abundances above 1 x $10^5$ cells $l^{-1}$ were observed, except between Sep-Nov in the Indian Ocean ($\approx 3.33$ x $10^4$ cells $l^{-1}$), Sep-Nov and Dec-Feb in the Atlantic Ocean ($\approx 5.40$ x $10^4$ and $\approx 9.78$ x $10^4$ cells $l^{-1}$ respectively) and Mar-May in the Pacific Ocean ($\approx 4.96$x $10^4$ cells $l^{-1}$).*

*The regions and periods with the highest mean heterococcolithophore abundance differ from the regions/periods with the highest maximum heterococcolithophore abundance. For example, the highest mean abundance is observed in the Indian Ocean during Mar-May ($\approx 1.13$ x $10^5$ ($\pm$ 2.97 x $10^4$) cells $l^{-1}$); which is higher than the highest mean abundance in the Southern Ocean observed during Mar-May ($\approx 1.17$ x $10^5$ ($\pm$ 2.88 x $10^4$) cells $l^{-1}$), and in the Arctic Circle during Jun-Aug ($\approx 5.83$ x $10^4$ ($\pm$ 2.97 x $10^4$) cells $l^{-1}$).*

*Although holococcolithophores show low abundances in the high latitudes of the Southern Hemisphere, highest maximum holococcolithophore abundances are observed in the Arctic circle ($>66°$ N) during Jun-Aug ($\approx 2.23$ x $10^5$ cells $l^{-1}$). High maximum abundances are additionally observed in the Mediterranean Sea (Sep-Nov) ($\approx 1.27$ x $10^5$ cells $l^{-1}$).*

*The lowest maximum holococcolithophore abundance is observed in the Pacific Ocean during Jun-Aug (4.45 x $10^2$ cells $l^{-1}$) and in the Arctic Circle during Sep-Nov (1.12 x $10^3$ cells $l^{-1}$).*

*On average, the Mediterranean Sea has the highest mean holococcolithophore abundance (between $\approx 2.21$ x $10^3$ and 9.42 x $10^3$ cells $l^{-1}$), followed by the Indian Ocean ($\approx 1.41$ x $10^3$ - 4.80 x $10^3$ cells $l^{-1}$). The lowest mean abundances are observed in the Pacific Ocean (4.9*

*x $10^1$ ($\pm$ 9.70 x $10^1$), Arctic Circle (Sep-Nov; 2.55 x $10^2$ ($\pm$ 2.71 x $10^2$) and Southern Ocean (Dec-Feb; $\approx$3.24 x $10^2$ ($\pm$ 2.06 x $10^2$) cells $l^{-1}$).*

*Depending on the season holococcolithophore contribution to total coccolithophore abundance varies globally between 1.67 % ($\pm$ 0.37%) in Dec-Feb and 16.16 % ($\pm$ 1.68%) in Jun-Aug, with highest contribution observed in the Mediterranean Sea in Jun-Aug (31.38 % $\pm$ 2.93 %) (Table 3). On an regional scale outside of the Mediterranean Sea, holococcolithophores contribute less than 8 % to the total coccolithophore abundances. However, the contribution of holococcolithophores to paired species is higher than when all hetero- and holococcolithophores are considered (Table 4), with a HOLP-index between 5.65 % ($\pm$1.71 %) and 27.41 ($\pm$2.67%) globally depending on season. The lowest HOLP-indices were observed in the Atlantic Ocean (Sep-Nov) (0.59 $\pm$0.81) and Dec-Feb (0.47 $\pm$0.65%), and in the Southern Ocean (Dec-Feb) (0.61 $\pm$0.58%). The highest HOLP-index was observed in the Mediterranean Sea in between Jun-Aug (39.03 $\pm$3.23%).*

Second, I do follow the nice overlap and nice expansions in the hyperspace. It is not clear to me how the authors transitioned from hypervolume to the nice space and how are the two haplo-diplontic stages represented in the Eq 2 (line 15) based on the similarity metrics? How was the intersection or the union between two hypervolumes determined?

We have updated the text to clarify that the hypervolume and niche space are the same thing [line 179, page 6]:

> *This hypervolume is considered to be the species niche space (Hutchinson, 1957) and allows niche comparisons between multiple phytoplankton - in this instance the two life cycle phases of coccolithophores.*

We have furthermore added Fig. 3, which illustrates the niche metrics utilized in this study, as well as a description of the figure [line 191, page 7]:

> *The niche metrics utilized in this study are illustrated in Fig. 3. Although we visualize the niche space of each species using contours, in reality the niche metrics are calculated based on random points sampled from the inferred hypervolumes (Blonder et al., 2014)*

Why was NE not calculated for BATS?
We have now included NE analysis of BATS [line 100, page 4]:

> *For the BATS, AMT and Mediterranean case studies, we additionally compiled temperature, salinity, and concentrations of DIN (nitrite + nitrate), phosphate, and silicate. For the AMT environmental variables were acquired from the British Oceanographic Data Centre (BODC). For the BATS environmental variables were acquired from the Bermuda Institute of Ocean Sciences (BIOS). For*

*the Mediterranean study, day length was calculated using the MIT Skyfield package in Python.*

[line 194, page 7]:

*We calculated the Jaccard and Sørensen-Dice similarity metrics and niche expansion for the AMT, BATS and Mediterranean Sea data sets. For the AMT data set, DIN showed high Pearson correlation to silicate ($\rho = 0.95$, $p < 0.001$) and phosphate ($\rho = 0.90$, $p < 0.001$). We thus only considered temperature, salinity and the concentration of DIN in this region. Although no such correlations were observed for the Mediterranean data set, and weaker but significant relationships were observed in the BATS stations ($\rho = 0.74$, $p < 0.001$ for silicate and $\rho = 0.84$, $p < 0.001$ for phosphate), to make the niche metrics comparable in all regions, the silicate and phosphate concentration of the Mediterranean and BATS data set were also excluded.*

[line 344, page 10]:

*We conducted niche similarity and niche expansion calculations on both the AMT, BATS and Mediterranean data sets to quantify niche space in these regions.*

[line 354, page 12]:

*For BATS, the niche overlap values are generally smaller than for the AMT, with a Jaccard overlap and Sørensen-Dice values of 0.60 and 0.66, respectively. The niche expansion of heterococcolithophores at BATS is larger compared to the AMT (0.49 versus 0.11 for BATS and AMT, respectively). The NE of holococcolithophores is similar for both stations (0.02 versus 0.05 for BATS and the AMT, respectively). S. anthos and S. pulchra are the only species for which coccolithophore life cycle pairs are observed at the BATS station. For these species, the NE of heterococcolithophore is similar to when all species were considered, but is higher for holococcolithophores. In the Mediterranean Sea, the niche overlap and niche expansion values are more similar to the BATS data set than to the AMT data set.*

line [372, page 12]:

*These results additionally suggest that the niche expansion patterns of the coccolithophore life cycle are more similar between the BATS and Mediterranean Sea than BATS and the AMT. This suggest that seasonal variations play an important role in structuring the niche space of coccolithophores, otherwise BATS and the AMT should be more alike due to more similar hydro-graphic conditions. This result highlights the value of time series for studying the ecology of the coccolithophore life cycle and raises the need for caution when comparing niche space of cruise data and time series.*

Third, in the section of seasonality (line 135), only a handful of environmental parameters are missing and there could be other important physical-chemical drivers. For example, In Figure 8, you also include turbulence in there, why was such parameter not included in the PCA analyses? What about pH, for example?

We have added a note explaining that other parameters influence the distribution patterns and niche expansion metrics of coccolithophores but are not included in our analysis [line 103, page 4]:

> *Other environmental variables such as turbulence, irradiance and pH might also impact coccolithophore distribution patterns, but we have not included them into our compilation because they are not available for all presented case studies.*

[line 200, page 7]:

> *It is likely however that silicate and phosphate as well as other parameters (such as irradiance, turbulence and carbonate chemistry) influence the niche space of coccolithophores and thus the metrics calculated. Beside the influence of environmental parameter choice, results of the niche space analysis will depend on what is considered a paired species. Although we use up to date definitions from Frada et al. (2019), these definitions are likely to change in the future. Finally, cryptic speciation (Geisen et al., 2002) and subsequently the pairing of multiple HOL phases to single HET phases and vice versa complicates results.*

We have also added some clarifications in our Margalef discussion to clarify that turbulence is not explicitly represented in our analysis [line 421, page 14]:

> *In terms of the ecological niche space - which is the environmental range a species inhabits - observations of hetero- and holo-coccolithophores in our meta-analysis broadly conform to the Margalef Niche Space Model. This model was proposed by Margalef (1978), and posits that the distribution of phytoplankton functional groups relate broadly to turbulence, light and nutrients. Although we do not explicitly represent the former, turbulence is implicitly represented in our analysis based on mixed-layer depth. Within the Margalef Niche Space framework, we find that hetero- and holo-coccolithophores occupy an intermediate functional group located between diatoms and dinoflagellates (see Fig. 9) as proposed by Houdan et al. (2006) and Frada et al. (2018).*

Also, as described are large-scale patterns, what about mesoscale type events, advection and other physical parameters?

We have now noted the potential influence of physical influences [line 137, page 5]:

*For more absolute estimates, additional sampling will have to be conducted. Although inter-annual variability and strong links between coincident climate variability and primary productivity (Behrenfeld et al., 2006) as well as inter-annual and mesoscale variability on local scales will influence phytoplankton distribution patterns (Volpe et al., 2012) which makes estimating global abundances challenging.*

In line 210, why were not the same approaches used for the Med and ATM? By using water column vs niche space approach, this excludes the possibility of comparing two regions.

We used different approaches for the two different data sets due to the different nature of the two data sets (AMT is spatial and Med is temporal). We have now clarified this and added that the variables included in the PCA will influence results. We have also included a figure with a reduced number of dimensions as a supplementary figure (Fig. S2-S4) [line 169, page 6]:

*Two different strategies were used to visualize the AMT and Mediterranean data sets as the AMT data set is spatial and the Mediterranean data set is temporal.*

[line 304, page 10]:

*The pattern observed in the PCA niche space should be interpreted with some caution because only a portion of the variance is captured (53%) and the use of interpolation introduces additional uncertainties. Besides, the structure of the PCA depends highly on the number and type of variables included (Fig. S2), in particular when time is considered.*

Forth, where is 17% of expanding niche space coming from?

We have now included a better description of how the niche expansion was calculated [line 205, page 7]:

*The environmental data were normalized using z-scores prior to analysis. Niche overlap and niche expansion were calculated only for species for which both life cycle phases were observed. In addition to calculating the niche expansion for individual species, we calculated a average niche expansion by taking the mean NE values of the grouped species for each location for both the haploid and diploid coccolithophores life cycle phases.*

Fifth, the authors report 7.3 to 18% of the species abundance, which is a relatively wide range and needs to be better quantified with the uncertainty.

These are two different numbers. We have now updated the discussion and conclusion to clarify this [line 404, page 13]:

*Our meta-analysis shows that holococcolithophores are a minor contributor to coccolithophore abundance in the modern ocean, contributing between ≈2-15 % to the total coccolithophore abundance*

*and between ≈5-30% of the total paired coccolithophore abundance depending on season.*

[line 507, page 16]:

*Our analysis shows that holococcolithophores constitute a minor proportion of total coccolithophore abundance (≈2-15 %), and constitute about ≈5-30% of total paired coccolithophore abundance depending on season.*

Also, given the quantitative estimates presented, I wish the authors to have better addressed some of the knowledge gaps, the impact on the pump of the haploid-diploid stage, standardizing the approaches to represent different species (paired- non paired, etc),.

We have now added some discussion on potential effects on the carbon pump as well as future climate change scenarios [line 410, page 12]:

*Although holococcolithophore contribution to calcium carbonate production is likely small due to their lower cellular $CaCO_3$ content - which is an order of magnitude lower than heterococcolithophores (Daniels et al., 2016; Fiorini et al., 2011a,b) - their role in the carbonate cycle in present, past and future oceans could have other biogeochemical effects. A shift towards a higher proportion of holococcolithophore cells, would result in lower global calcium carbonate production which could subsequently result in lower $CO_2$ outgassing on short time scales. Furthermore the ballasting effect of coccolithophores would be reduced if a shift towards more lightly calcified haploid cells occurred (Hoffmann et al., 2015) which would potentially reduce efficiency of the carbon pump by reducing sinking rates. Although how other factors such as shifts in carbonate chemistry impact holococcolithophore abundance are not clear, increased stratification and decreased nutrient supply are projected under the RCP 8.5 climate change scenario (Fu et al., 2016), which would favor holococcolithophores. This shift from diploid to haploid coccolithophores could on the one hand reduce $CO_2$ outgassing, but would additionally reduce ballasting and subsequently impact the carbon pump by reducing sinking rates.*

We have furthermore included our identified knowledge gaps in the abstract [line 15, page 1]:

*Our compilation highlights the spatial and temporal sparsity of SEM measurements, and the need for new molecular techniques to identify uncalcified haploid coccolithophores. Our work also emphasizes the need for further work on the carbonate chemistry niche space of the coccolithophore life cycle.*

Based in Figure 3, one could conclude that the relative abundances (f,g,h) of holococcos are only slightly lower compared to the heterococcos (c,d,e) and these figures need changes.

We have updated the axis to absolute abundance of Fig 3. (see Fig. 4 [page 28]).

What is the difference in shading?

The light shading on the latitudinal plots is log transformed (as noted in the caption). We have updated the axis to better reflect this (see updated Fig. 4 [page 28]).

**BG Discussion: Reply to RC3**

We would like to thank the third reviewer for their positive and constructive feedback. Our in-depth response can be found below.

General comments This is an interesting paper, timely, and relevant to the field of physiological ecology of phytoplankton. It deserves to be published but needs some minor revision. The paper was a bit sloppy in spots, with a number of typos. The paper should be checked over carefully prior to final submission.There are some terms that really need to be clarified in the revision to avoid confusion and to sharpen their points. First, when discussing nitrogenous nutrients they refer to "fixed nitrogen" as the sum of nitrate and nitrite (line 90-91). This reviewer has no idea why they are using the adjective "fixed" for the sum of these molecules (and they do not include ammonium or urea in that sum, for example). Typically, the fixation of nitrogen by phytoplankton is describing the uptake and assimilation of N2 gas into organic nitrogen fractions, which is not what they are describing. I would advocate that they globally scrub the term "fixed nitrogen" and replace it with something like dissolved inorganic nitrogen (DIN, here defined as nitrate + nitrite only).

We have replaced every instance of 'Fixed nitrogen' with dissolved inorganic nitrogen (DIN) as suggested.

Second, in their equation about niche expansion (line 150) they refer to terms describing the "intersection of hypervolumes" and the "union of hypervolumes". If there is a union of hypervolumes, then they also intersect, right? The authors must very carefully define the difference between these. As long as there are ambiguities in the definition of those terms, then the entire niche expansion argument won't have much relevance.

The reviewer points out to potential for confusion between the intersection and union of hypervolumes. In set theory, the union includes all data points. While the difference is only the overlapping set of data points. We have now included Figure 3 as well as description of the figure to clarify this [line 191, page 7]:

> The niche metrics utilized in this study are illustrated in Figure 3. Although we visualize the niche space of each species using contours, in reality the niche metrics are calculated based on random points sampled from the inferred hypervolumes (Blonder et al., 2014)

Finally, they talk about a 7% contribution of holococcolithophore abundance to the total coccolithophore abundance as being significant (abstract and line 331). It may be statistically significant, but it seems to this reviewer to be a little overblown. I would suggest that holococcolithophores more appropriately would be considered a minor constituent of the total coccolithophore assemblage. For holo/heterococcolith paired species, the holococcolithophore abundance represents ≈18% of the paired species abundance, only about a fifth, definitely still a minor fraction, at best. This doesn't detract from the results. It is still a fascinating observation and the question that arises to this reviewer is why is that fraction so small?

We have updated the text to clarify that holococcolithophores constitute a minor component of total coccolithophore abundance but are important in terms of niche space [line 1, page 1]:

> *Coccolithophores are globally important marine calcifying phytoplankton that utilize a haplo-diplontic life cycle. The haplo-diplontic life cycle allows coccolithophores to divide in both life cycle phases, and has been proposed to allow coccolithophores to expand their niche space. To-date research has, however, largely overlooked the life cycle of coccolithophores, and instead focused on the diploid life cycle phase of coccolithophores. Through the synthesis and analysis of global scanning electron microscopy (SEM) coccolithophore abundance data (n = 2534), we find that calcified haploid coccolithophores generally constitute a minor component of the total coccolithophore abundance ($\approx 2-15\%$ depending on season). However, using case studies in the Atlantic Ocean and Mediterranean Sea, we show that depending on environmental conditions calcifying haploid coccolithophores can can be significant contributors to the coccolithophore standing stock (up to $\approx 30\%$). Furthermore, using hypervolumes to quantify the niche space of coccolithophores, we illustrate that the haploid and diploid life cycle phases inhabit contrasting niches, and that on average, this allows coccolithophores to expand their niche space by $\approx 18.8$ % with a range of 3-76% for individual species.*

[line 404, page 13]:

> *Our meta-analysis shows that holococcolithophores are a minor contributor to coccolithophore abundance in the modern ocean, contributing between $\approx 2-15$ % to the total coccolithophore abundance and between $\approx 5-30\%$ of the total paired coccolithophore abundance depending on season. However, our analysis also shows that haploid cells play an important role in coccolithophore ecology, accounting for $\approx 19\%$ of their niche space, which lesser or greater contributions depending on the species (3-76%). Our analysis furthermore shows that if conditions are favorable (specifically increased stratification and reduced nutrient supply) holococcolithophores can be significant contributors to the coccolithophore standing stock (up to $\approx 30\%$).*

*Our analysis shows that holococcolithophores constitute a minor proportion of total coccolithophore abundance (≈2-15 %), and constitute about ≈5-30% of total paired coccolithophore abundance depending on season.*

Additionally, we have added some discussion as to why the contribution of holococcolithophores is so small [line 475, page 15]:

*Nonetheless, from our compilation it is clear that holococcolithophores constitute a minor component of total coccolithophore abundance. This could be in part to sampling bias, specifically temporal bias to periods of high heterococcolithophore abundance. However, other factors such as the strong dominance of E. huxleyi which has a naked haploid phase, and the limited biomass low nutrient regions are able to sustain might also exert a significant influence. The low contribution of holococcolithophores is interesting and raises the question what physiological traits make heterococcolithophores generally more successful in the modern ocean.*

This paper requires some revision but it provides new insights to a very real problem in coccolithophore ecology. It deserves to be published and will be cited well. The authors simply need to clean it up a bit.

Specific comments

Line 4 after "diploid life cycle phase" are they referring to coccolithophores only or other organisms. Please clarify.

done, [line 4, page 1]:

*To-date research has however largely overlooked the life cycle of coccolithophores, and has instead focused on the diploid life cycle phase of coccolithophores.*

Line 13 "ballast" not "ballasts"

done, [line 20, page 2]:

Line 13-15 They are describing the biological carbon pump, not the carbonate pump(aka alkalinity pump). The linkage of calcite production to the biological carbon pump is a strong one via ballasting of organic carbon to the sea floor. This is not the alkalinity pump however. Klass and Archer (2002) were looking at the impact of ballasting of sinking POC and the effect on the rain ratio.

We have have clarified this and added a reference [line 20, page 2]:

*Coccoliths eventually rain down into the ocean interior or serve as ballasts as they are incorporated into faecal pellets and aggregates, which drives the carbonate pump and enhances the organic carbon pump by increasing organic carbon export rates to the deep sea (Klaas and Archer, 2002; Zeebe, 2012).*

Line 16 Globally, about one quarter of all marine sediments are calcium carbonate. Citing the 30-90% value presents a skewed view of the importance of calcite sediments on Earth.

We present the 30-90% values to highlight the importance of coccolithophores in accounting for calcium carbonate sediment burial (and not the importance of calcium carbonate in sediments. We have adjusted our sentence to avoid any confusion [line 22, page 2]:

> *Through the production of coccoliths, coccolithophores produce ≈1.5 Pg of inorganic carbon per year (Hopkins and Balch, 2018; Krumhardt et al., 2019), and subsequently account for 30 % to 90 % of carbonate in sediments (Broecker and Clark, 2009), highlighting the importance of coccolithophores in calcium carbonate burial.*

Line 18- Given that this sentence is going back to the biological carbon pump, you might move it up in the paragraph where you are first mentioning the biological carbon pump.

Line 30 add an "s"..."A few organism"

done [line 38, page 2]

Line 91 Reference to "fixed nitrogen" and all subsequent uses of that term in the paper...see general comments above.

done (see above)

Lines 151 and 152- Must describe how the "union" and "intersection" of hypervolumes are being distinguished. See general comments above.

done (see above)

Line 162- Again, they are describing the function in R to calculate the "intersection" of hypervolumes when the reader may not be clear about the difference between this calculation and that of the union of the hypervolumes! This is a really important distinction.

done (see above)

Line 185 As they state, on a regional basis, holococcolithophores generally contributed <6% of total coccolithophore abundance. This seems pretty minor to be honest!

done (see above)

Line 190- change to..."where a HOLP-index"...not "an Holp-index"

This sentence was removed.

Line 193- add comma, "In the global data set, heterococcolithophore..."

done, [line 245, page 7]

Line 218- change to..."high nutrient concentrations, cold water temperatures at depth or other factors not addressed in this study".

done, [line 270, page 9]

Line 253 They show significant positive correlations with silicate. This is a very interesting observation. The Discussion section should have a few sentences explaining how this could be!

We have now added the following discussion [line 325, page 11]:

*In the Mediterranean Sea and the Atlantic Ocean significant negative correlations were observed between holococcolithophores and silicate. Although this correlation could be in part to strong correlation between DIN and Silicate ($\rho$= 0.95) observed in the Atlantic Ocean, the reason for this is less clear in the Mediterranean Sea as no such correlation is observed. A physiological reason for the negative correlation to silicate could be different silicate requirements among different coccolithophore species. Durak et al. (2016) for instance found evidence of silicate requirement for the heterococcolith life cycle phases of S. apsteinii, C. coccolithus and C. leptoporus but not for E. huxleyi or G. oceanica. Follow up experiments have furthermore found holococcolith life cycle phases of C. coccolithus and C. leptoporus do not require silicate (manuscript by Langer et al., in prep).*

Line 273- add "s" to heterococcolithophore to make it plural.
done, [line 336, page 11]
Line 285 add comma, "overlap metrics, respectively" done, [line 350, page 12]

Line 331 "Our meta-analysis shows that holococcolithophores are important contributors to coccolithophore abundance and ecology contributing $\approx$7.3% to total coccolithophore abundance" This observation doesn't match the data. 7.3% is a small number. Call it like it is!
done (see above)

Line 336- re-word this so that it agrees with the minor contribution..."past and future oceans could have other biogeochemical effects. A shift towards"...
done, [line 412, page 12]
Line 363- Remove "?"
Our bibliography was missing an BibTeX entry. This is now fixed. [line 446, page 15]
Line 366- improper hyphenation of wrap-around word "coccolithophore"
Line 370- I disagree with this statement. Calcification measurements are including the calcite production of holo- and heterococcoliths. However, the standing stock of calcite is being underestimated by not including the holococcolithophore abundance. Also, leave off the last words of the sentence, "or activity".
We have updated the section to reflect this [line 453, page 15]:

*Secondly, we underestimate coccolithophore primary productivity and calcite standing stock by not including accurate assessments of their abundance.*

Line 380- Sentence "Overall observations in the haploid stage of E. huxleyi are...".There is some classic literature that the authors should cite from the mid 1990's: Campbell, L., et al. (1994). "Immunochemical characterization for eukaryotic ultraplankton from the Atlantic and Pacific oceans." Journal

of Plankton Research 16(1): 35-51. They used immunochemical antisera to identify haploid stages of E. huxleyi.

We have now included the Campbell et al. (1994) reference [line 464, page 15]:

> *Overall, observations in the haploid stage of E. huxleyi are extremely limited due to difficulty of identifying the haploid phase with regular light microscopy, highlighting the need for developing new techniques to account for this potentially important life cycle stage. Further development of FISH (Campbell et al., 1994) and COD-FISH (Frada et al., 2012) methodologies would be particularly relevant in this context.*

Line 385;Again, there were a number of classic SEM studies from the Pacific Ocean. One by Reid (1980). Reid, F. (1980). "Coccolithophorids of the North Pacific Central Gyre with notes on their vertical and seasonal distribution." Micropaleontology 26: 151-176. The SEM plates in the paper are meticulous and it might be worth a look before you discount all Pacific SEM observations. See also previous work of Honjo and Okada from the Pacific.

We have updated our methods section and discussion to reflect this.
[line 88, page 2]:

> *Since abundance data was manually compiled, our data set is not exhaustive. For instance, some SEM studies such as those by Okada and Honjo (1973); Honjo and Okada (1974); Reid (1980) are not included in this data set since the data were not retrievable from the original publications.*

[line 469, page 15]:

> *There is for instance a lack of SEM observations in the Pacific Ocean (2 studies in this compilation). Although, this is in part because existing data from the 1980s was not retrievable, and because the low abundance of coccolithophores in this region means that it has been of low priority for time-intensive and costly re-sampling. In addition, there is a limited number of available SEM time series, which are particularly valuable due to the seasonal nature of these organisms and the importance of time in structuring coccolithophore niche space.*

Line 404- There is contrary evidence you should cite to be balance, though: Rivero-Calle, S., et al. (2015). "Multidecadal increase in North Atlantic coccolithophores and the potential role of rising CO2." Science 350(6267): 1533-1537. done [line 499, page 16]:

> *Furthermore, contradicting evidence suggesting increased coccolithophore abundance in response to higher has been noted in situ (Rivero-Calle et al., 2015)*

Line 407- eliminate the "s" from "compositions"

done [line 499, page 16]

Line 414- Reword, "Our analysis shows that holococcolithophores constitute about one fifth of total paired coccolithophore abundance..."

We have kept the number as a percentage for consistency.

Figures:

Figures 3 and 4- The font on all the axes is way to small to be readable. These must be increased in size.

done, although we would like to note that high resolution scalable images will be provided with the final manuscript [page 28, page 29]

Figure 4- Legend is reversed for red and blue colors...Heterococcos are plotted in red(not blue) and holococcos are plotted in blue (not red).

done [page30]

Figures 5 and 6- No units are provide in this figure or the legend for the color bars!

done [page 31, 32]

Fig. 6 change "fixed nitrogen" to DIN (see also Fig. 7)

done [page 33]

Table 2 is excellent and a great reference. Should you state the names for the holoforms of R. clavigera and R. xiphos since you have left them blank?

They are blank because *A. robusta*, *R. clavigera*, and *R. xiphos* are all associated with *S. quadridentata*. The same is true for *S. pulchra* and *S. protrudens* which are both associated with *S. pulchra* HOL. We have updated the table to reflect this [page 35].

Tables 6 and 7- You never discuss the significant relationships with Silicate (not "Si" as you say in the table!) This really deserves some discussion.

done (see above)

Tables 8 and 9 The legends are very minimalistic. Please move your definition of NE1and NE2 to the legend from the footnotes. This needs to be more obvious to the reader. Also, maybe specify in the table legend what the Jaccard and Sorensen columns refer to (and units?) or refer the reader to the text.

We have replace NE1 and NE2 with NE HET and NE HOL, and refer the reader to the text and Fig. 3 for definitions. We additionally merged Table 8 and 9 [page 41].

[revised manuscript text omitted]
 ($\pm$ci) (cells L$^{-1}$) | max (cells L$^{-1}$) | contribution ($\pm$ci)(%) | n |
|---|---|---|---|---|---|---|
| Global | Mar-May | HET | 4.57e+04 ($\pm$4.72e+03) | 4.93e+05 | 93.72 ($\pm$0.98) | 585 |
| | | HOL | 2.00e+03 ($\pm$5.42e+02) | 8.72e+04 | 5.05 ($\pm$0.93) | 585 |
| | Jun-Aug | HET | 4.36e+04 ($\pm$1.56e+04) | 4.37e+06 | 82.53 ($\pm$1.68) | 739 |
| | | HOL | 4.64e+03 ($\pm$1.03e+03) | 2.23e+05 | 16.16 ($\pm$1.68) | 739 |
| | Sep-Nov | HET | 1.75e+04 ($\pm$3.09e+03) | 3.53e+05 | 91.46 ($\pm$1.61) | 438 |
| | | HOL | 1.74e+03 ($\pm$8.27e+02) | 1.27e+05 | 7.11 ($\pm$1.59) | 438 |
| | Dec-Feb | HET | 9.32e+04 ($\pm$8.99e+03) | 1.64e+06 | 95.37 ($\pm$0.66) | 772 |
| | | HOL | 1.78e+03 ($\pm$4.76e+02) | 1.18e+05 | 1.67 ($\pm$0.37) | 772 |
| Arctic Circle | Jun-Aug | HET | 5.83e+04 ($\pm$4.53e+04) | 4.37e+06 | 95.87 ($\pm$2.12) | 213 |
| | | HOL | 1.83e+03 ($\pm$2.18e+03) | 2.23e+05 | 3.71 ($\pm$2.02) | 213 |
| | Sep-Nov | HET | 3.41e+04 ($\pm$2.51e+04) | 1.29e+05 | 94.79 ($\pm$5.53) | 11 |
| | | HOL | 2.55e+02 ($\pm$2.71e+02) | 1.12e+03 | 5.21 ($\pm$5.53) | 11 |
| East China Sea | Sep-Nov | HET | 2.99e+04 ($\pm$1.30e+04) | 2.39e+05 | 96.48 ($\pm$3.97) | 51 |
| | | HOL | 9.06e+02 ($\pm$7.98e+02) | 1.47e+04 | 3.52 ($\pm$3.97) | 51 |
| Indian Ocean | Mar-May | HET | 1.13e+05 ($\pm$2.97e+04) | 2.18e+05 | 96.88 ($\pm$1.3) | 33 |
| | | HOL | 1.41e+03 ($\pm$7.85e+02) | 1.10e+04 | 2.35 ($\pm$1.31) | 33 |
| | Jun-Aug | HET | 2.40e+04 ($\pm$7.38e+03) | 1.11e+05 | 90.11 ($\pm$3.53) | 53 |
| | | HOL | 6.57e+02 ($\pm$2.67e+02) | 3.43e+03 | 3.68 ($\pm$2.09) | 53 |
| | Sep-Nov | HET | 7.03e+03 ($\pm$1.50e+03) | 3.33e+04 | 89.33 ($\pm$3.78) | 89 |
| | | HOL | 2.87e+02 ($\pm$2.00e+02) | 5.63e+03 | 5.57 ($\pm$3.49) | 89 |
| | Dec-Feb | HET | 2.00e+05 ($\pm$1.71e+03) | 2.27e+05 | 96.56 ($\pm$0.64) | 102 |
| | | HOL | 4.80e+03 ($\pm$1.27e+03) | 3.10e+04 | 2.3 ($\pm$0.6) | 102 |
| Mediterranean Sea | Mar-May | HET | 2.80e+04 ($\pm$4.90e+03) | 2.11e+05 | 88.88 ($\pm$3.19) | 146 |
| | | HOL | 3.76e+03 ($\pm$1.83e+03) | 8.72e+04 | 10.55 ($\pm$3.06) | 146 |
| | Jun-Aug | HET | 1.21e+04 ($\pm$1.95e+03) | 1.00e+05 | 68.42 ($\pm$2.92) | 290 |
| | | HOL | 9.42e+03 ($\pm$1.94e+04) | 1.02e+05 | 31.38 ($\pm$2.93) | 290 |
| | Sep-Nov | HET | 1.70e+04 ($\pm$4.38e+03) | 3.53e+05 | 89.11 ($\pm$2.89) | 195 |
| | | HOL | 3.10e+03 ($\pm$1.82e+03) | 1.27e+05 | 10.2 ($\pm$2.91) | 195 |
| | Dec-Feb | HET | 4.23e+04 ($\pm$1.18e+04) | 3.96e+05 | 96.78 ($\pm$1.11) | 125 |
| | | HOL | 2.21e+03 ($\pm$2.33e+03) | 1.18e+05 | 1.96 ($\pm$0.92) | 125 |
| Atlantic Ocean | Mar-May | HET | 4.20e+04 ($\pm$5.68e+03) | 1.83e+05 | 96.78 ($\pm$0.87) | 174 |
| | | HOL | 1.20e+03 ($\pm$5.36e+02) | 2.76e+04 | 1.88 ($\pm$0.6) | 174 |
| | Jun-Aug | HET | 1.51e+05 ($\pm$6.84e+04) | 1.55e+06 | 93.6 ($\pm$2.07) | 86 |
| | | HOL | 1.70e+03 ($\pm$8.96e+02) | 2.29e+04 | 3.89 ($\pm$1.6) | 86 |
| | Sep-Nov | HET | 1.48e+04 ($\pm$4.56e+03) | 5.40e+04 | 96.76 ($\pm$1.34) | 30 |
| | | HOL | 3.77e+02 ($\pm$1.54e+02) | 1.39e+03 | 2.59 ($\pm$1.22) | 30 |
| | Dec-Feb | HET | 2.50e+04 ($\pm$9.70e+03) | 9.78e+04 | 94.23 ($\pm$3.09) | 29 |
| | | HOL | 3.38e+02 ($\pm$1.55e+02) | 1.64e+03 | 1.81 ($\pm$1.26) | 29 |
| Pacific Ocean | Mar-May | HET | 1.43e+04 ($\pm$6.24e+03) | 4.96e+04 | 92.33 ($\pm$6.63) | 25 |
| | | HOL | 4.50e+03 ($\pm$4.56e+03) | 3.98e+04 | 7.55 ($\pm$6.64) | 25 |
| | Jun-Aug | HET | 1.96e+04 ($\pm$3.17e+04) | 1.48e+05 | 98.12 ($\pm$1.64) | 9 |
| | | HOL | 4.90e+01 ($\pm$9.70e+01) | 4.45e+02 | 0.03 ($\pm$0.07) | 9 |
| | Dec-Feb | HET | 2.00e+04 ($\pm$1.32e+04) | 1.64e+05 | 94.85 ($\pm$4.02) | 28 |
| | | HOL | 8.65e+02 ($\pm$1.20e+03) | 1.70e+04 | 0.89 ($\pm$0.75) | 28 |
| Southern Ocean | Mar-May | HET | 1.17e+05 ($\pm$2.88e+04) | 4.93e+05 | 98.19 ($\pm$0.88) | 50 |
| | | HOL | 9.10e+02 ($\pm$6.55e+02) | 1.60e+04 | 1.36 ($\pm$0.87) | 50 |
| | Dec-Feb | HET | 9.10e+04 ($\pm$1.66e+04) | 1.64e+06 | 99.05 ($\pm$0.7) | 332 |
| | | HOL | 3.24e+02 ($\pm$2.06e+02) | 2.67e+04 | 0.95 ($\pm$0.7) | 332 |

Values in parentheses are 95% confidence intervals.

**Table 4.** Global HOLP-index

[revised manuscript text omitted]